# Gaussian Loss Smoothing Enables Certified Training with Tight Convex Relaxations

**Stefan Balauca**  *stefann.balauca@gmail.com*
*INSAIT, Sofia University "St. Kliment Ohridski", Bulgaria*

**Mark Niklas Müller**  *mark.niklas.mueller@gmail.com*
*LogicStar.ai*
*(work done while at ETH Zürich)*

**Yuhao Mao**  *yuhao.mao@inf.ethz.ch*
*Department of Computer Science, ETH Zürich, Switzerland*

**Maximilian Baader**  *mbaader@inf.ethz.ch*
*Department of Computer Science, ETH Zürich, Switzerland*

**Marc Fischer**  *marc.fischer@inf.ethz.ch*
*Invariant Labs*
*(work done while at ETH Zürich)*

**Martin Vechev**  *martin.vechev@inf.ethz.ch*
*Department of Computer Science, ETH Zürich, Switzerland*

**Reviewed on OpenReview:** *https://openreview.net/forum?id=lknvxcjuos*

## Abstract

Training neural networks with high certified accuracy against adversarial examples remains an open challenge despite significant efforts. While certification methods can effectively leverage tight convex relaxations for bound computation, in training, these methods, perhaps surprisingly, can perform worse than looser relaxations. Prior work hypothesized that this phenomenon is caused by the discontinuity, non-smoothness, and perturbation sensitivity of the loss surface induced by tighter relaxations. In this work, we theoretically show that applying Gaussian Loss Smoothing (GLS) on the loss surface can alleviate these issues. We confirm this empirically by instantiating GLS with two variants: a zeroth-order optimization algorithm, called PGPE, which allows training with non-differentiable relaxations, and a first-order optimization algorithm, called RGS, which requires gradients of the relaxation but is much more efficient than PGPE. Extensive experiments show that when combined with tight relaxations, these methods surpass state-of-the-art methods when training on the same network architecture for many settings. Our results clearly demonstrate the promise of Gaussian Loss Smoothing for training certifiably robust neural networks and pave a path towards leveraging tighter relaxations for certified training.

## 1 Introduction

The increased deployment of deep learning systems in mission-critical applications has made their provable trustworthiness and robustness against adversarial examples (Biggio et al., 2013; Szegedy et al., 2014) an important topic. As state-of-the-art neural network certification has converged to similar approaches (Zhang et al., 2022; Ferrari et al., 2022), increasingly reducing the verification gap, the focus in the field is now shifting to specialized training methods that yield networks with high certified robustness while minimizing the loss of standard accuracy (Müller et al., 2023; Mao et al., 2023a; De Palma et al., 2024).

| Relaxation | Tightness | GRAD [%] | | PGPE [%] | RGS [%] |
|:----------:|:---------:|:--------:|:--:|:--------:|:-------:|
| IBP | 0.55 | 91.23 | Loss | 87.02 | 90.46 |
| CROWN-IBP | 1.68 | 88.76 | Smoothing | 90.23 | 90.71 |
| DEEPPOLY | 2.93 | 90.04 | $\Longrightarrow$ | 91.53 | 91.88 |

Figure 1: Illustration of how Gaussian loss smoothing enables certified training with tight relaxations. We compare the Certified Accuracy [%] obtained by training a `CNN3` network on MNIST $\epsilon = 0.1$ with different relaxations using either the standard gradient (GRAD, used as baseline) or a gradient estimate computed on the smoothed loss surface (PGPE and RGS) with the empirical tightness of the method. Using GRAD methods produces the best results with the least tight IBP relaxation, which is known as the paradox of certified training. However, using GLS-based optimization methods PGPE and RGS, we obtain a clear correlation between relaxation tightness and performance.

**Certified Training**  State-of-the-art (SOTA) certified training methods aim to optimize the network's worst-case loss over an input region defined by an adversarial specification. However, as computing the exact worst-case loss is NP-complete (Katz et al., 2017), they typically utilize convex relaxations to compute over-approximations of this loss (Gowal et al., 2018; Singh et al., 2018; 2019). Surprisingly, training methods based on the least precise relaxations (IBP) empirically yield the best performance (Shi et al., 2021), while tighter relaxations perform progressively worse (left, Figure 1). Jovanović et al. (2022) and (Lee et al., 2021) investigated this surprising phenomenon which they call the "Paradox of Certified Training", both theoretically and empirically, and found that tighter relaxations induce harder optimization problems. Specifically, they identify the *continuity*, *smoothness*, and *sensitivity* of the loss surface induced by a relaxation as key factors for the success of certified training, beyond its *tightness*. Indeed, *all* state-of-the-art methods are based on the imprecise but continuous, smooth, and insensitive IBP bounds (Müller et al., 2023; Mao et al., 2023a; De Palma et al., 2024). However, while these IBP-based methods improve robustness, they induce severe regularization, significantly limiting the effective capacity and thus standard accuracy (Mao et al., 2023b). This raises the following fundamental question:

*Can we enable certified training with tight convex relaxations by addressing the discontinuity, non-smoothness, and perturbation sensitivity, thus obtaining a better robustness-accuracy trade-off?*

**This Work: Enabling Certified Training with Tight Convex Relaxations**  In this work we propose a conceptual path forward to overcoming the paradox by addressing the three issues identified by prior works. Our key insight is that the discontinuity, non-smoothness, and perturbation sensitivity of the loss surface can be mitigated by smoothing the training loss with a Gaussian kernel over the network parameter search space. We refer to this approach as *Gaussian Loss Smoothing* (GLS). To instantiate GLS, we propose to augment certified training using two methods: (1) a gradient-free method based on Policy Gradients with Parameter-based Exploration (PGPE) (Sehnke et al., 2010) and (2) a gradient-based method based on Randomized Gradient Smoothing (RGS) (Starnes et al., 2023). While both methods approximate GLS which is intractable to compute exactly, they enjoy different benefits: (1) PGPE allows training with non-differentiable relaxations, while (2) RGS is much more efficient than PGPE. We note that the GLS principles, as well as the PGPE and RGS algorithms are independent of the training loss we choose to optimize, and therefore can be applied on top of any certified training method and any convex relaxation.

Using these GLS methods, we empirically demonstrate that tighter relaxations can indeed lead to strictly better networks, thereby confirming the importance of addressing discontinuity, non-smoothness, and perturbation sensitivity (right, Figure 1). Critically, with the more precise DEEPPOLY relaxation (Singh et al., 2019), we show that GLS methods achieve strictly better results than the less precise IBP. Moreover, we demonstrate that the advantages of GLS improve with increasing network depth, outperforming state-of-the-art methods applied for the same architecture in many settings, particularly when precision matters more. Our results demonstrate the promise of GLS for training certifiably robust neural networks and pave a path towards leveraging tighter relaxations for certified training.

**Main Contributions**   Our core contributions are:

1. A theoretical investigation showing Gaussian Loss Smoothing (GLS) mitigates discontinuity, non-smoothness, and perturbation sensitivity of the loss surface in certified training with tight relaxations.
2. A PGPE-based adaptation of certified training that approximates GLS in zeroth-order optimization, enabling training with tight non-differentiable relaxations.
3. A RGS-based adaptation of certified training that approximates GLS in first-order optimization, requiring differentiable relaxations, but achieving a speedup of up to 40x compared to PGPE.
4. A comprehensive empirical evaluation of different convex relaxations under GLS with the proposed methods, demonstrating the promise of GLS-based approaches.

## 2   Training for Certified Robustness

Below, we first introduce the setting of adversarial robustness before providing a background on (training for) certified robustness. For a detailed notation guide see §A.

### 2.1   Adversarial Robustness

We consider a neural network $\boldsymbol{f_\theta}(\boldsymbol{x})\colon \mathcal{X} \to \mathbb{R}^n$, parameterized by the weights $\boldsymbol{\theta}$, that assigns a score to each class $i \in \mathcal{Y}$ given an input $\boldsymbol{x} \in \mathcal{X}$. This induces the classifier $F\colon \mathcal{X} \to \mathcal{Y}$ as $F(\boldsymbol{x}) \coloneqq \arg\max_i \boldsymbol{f_\theta}(\boldsymbol{x})_i$. We call $F$ locally robust for an input $\boldsymbol{x} \in \mathcal{X}$ if it predicts the same class $y \in \mathcal{Y}$ for all inputs in an $\epsilon$-neighborhood $\mathcal{B}_p^\epsilon(x) \coloneqq \{\boldsymbol{x}' \in \mathcal{X} \mid \|\boldsymbol{x} - \boldsymbol{x}'\|_p \leq \epsilon\}$. To prove that a classifier is locally robust, we thus have to show that $F(\boldsymbol{x}') = F(\boldsymbol{x}) = y, \forall \boldsymbol{x}' \in \mathcal{B}_p^\epsilon(x)$.

**Adversarial Attacks and Empirical Robustness**   Disproving local robustness for a given input $\boldsymbol{x}$ is done by finding an *adversarial example* $\boldsymbol{x}' \in \mathcal{B}_p^\epsilon(x)$ such that $F(\boldsymbol{x}') \neq F(\boldsymbol{x})$. The procedure of searching for adversarial examples is called *adversarial attack*. The most common attack methods (Goodfellow et al., 2015; Madry et al., 2018) use first-order gradient information to maximize the loss function associated with $\boldsymbol{x}'$. When such a method fails to find an adversarial example, we say that the network is *empirically robust* for the given input $\boldsymbol{x}$ and perturbation radius $\epsilon$.

**Robustness Guarantees**   Local robustness is equivalent to the log-probability of the target class $y$ being greater than that of all other classes for all relevant inputs, i.e.,

$$\min_{\boldsymbol{x}' \in \mathcal{B}, i \neq y} f(\boldsymbol{x}')_y - f(\boldsymbol{x}')_i > 0. \tag{1}$$

As solving this neural network verification problem exactly is generally NP-complete (Katz et al., 2017), state-of-the-art neural network verifiers relax it to an efficiently solvable convex optimization problem (Brix et al., 2023). To this end, the non-linear activation functions are replaced with convex relaxations in their input-output space, allowing linear bounds of the following form on their output $f(\boldsymbol{x})$ to be computed:

$$\boldsymbol{A}_l \boldsymbol{x} + \boldsymbol{b}_l \leq \boldsymbol{f_\theta}(\boldsymbol{x}) \leq \boldsymbol{A}_u \boldsymbol{x} + \boldsymbol{b}_u, \tag{2}$$

for some input region $\mathcal{B}_p^\epsilon(x)$. From these symbolic bounds we can obtain concrete numerical bounds, usually in a layer-wise fashion, as $\boldsymbol{l}_j = \min_{\boldsymbol{x} \in \mathcal{B}} \boldsymbol{A}_{l_j} \boldsymbol{x} + \boldsymbol{b}_{l_j}$, and $\boldsymbol{u}_j$ analogously, for each layer $j$. Hence, for the last layer $m$ we obtain bounds for the score associated with each class $\boldsymbol{l}_m \leq \boldsymbol{f}(\boldsymbol{x}) \leq \boldsymbol{u}_m$, which can in turn be used to verify Equation (1).

To obtain (certifiably) robust neural networks, specialized training methods are required. For a data distribution $(\boldsymbol{x}, t) \sim \mathcal{D}$, standard training optimizes the network parametrization $\boldsymbol{\theta}$ to minimize the expected cross-entropy loss $\theta_{\text{std}} = \arg\min_\theta \mathbb{E}_\mathcal{D}[\mathcal{L}_{\text{CE}}(\boldsymbol{f_\theta}(\boldsymbol{x}), t)]$ with $\mathcal{L}_{\text{CE}}(\boldsymbol{y}, t) = \ln\left(1 + \sum_{i \neq t} \exp(y_i - y_t)\right)$. To train for robustness, we, instead, aim to minimize the expected *worst-case loss* for a given robustness specification, leading to a min-max optimization problem: $\theta_{\text{rob}} = \arg\min_\theta \mathbb{E}_\mathcal{D}\left[\max_{\boldsymbol{x}' \in \mathcal{B}^\epsilon(\boldsymbol{x})} \mathcal{L}_{\text{CE}}(\boldsymbol{f_\theta}(\boldsymbol{x}'), t)\right]$. As computing the worst-case loss by solving the inner maximization problem is generally intractable, it is commonly under- or over-approximated, yielding adversarial and certified training, respectively.

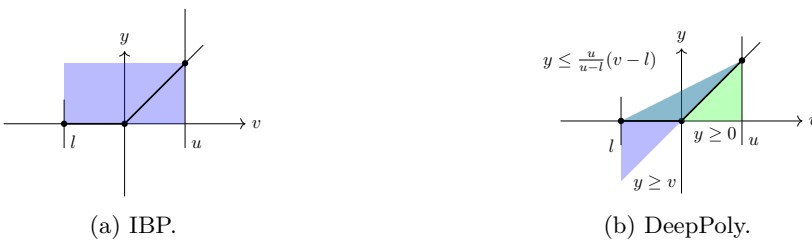

(a) IBP.  (b) DeepPoly.

Figure 2: IBP and DEEPPOLY relaxations of a ReLU with bounded inputs $v \in [l, u]$. For DEEPPOLY the lower-bound slope $\lambda$ is chosen to minimize the area between the upper and lower bounds in the input-output space, resulting in the blue or green area.

**Adversarial Training** optimizes a lower bound on the inner optimization objective. To this end, it first computes concrete examples $\boldsymbol{x}' \in \mathcal{B}^\epsilon(\boldsymbol{x})$ that approximately maximize the loss term $\mathcal{L}_{\mathrm{CE}}$ and then optimizes the network parameters $\boldsymbol{\theta}$ for these examples. While networks trained this way typically exhibit good empirical robustness, they remain hard to formally certify and are sometimes vulnerable to stronger attacks (Tramèr et al., 2020; Croce & Hein, 2020).

**Certified Training** typically optimizes an upper bound on the inner maximization objective. To this end, the robust cross-entropy loss $\mathcal{L}_{\mathrm{CE,rob}}(\mathcal{B}^\epsilon(\boldsymbol{x}), t) = \mathcal{L}_{\mathrm{CE}}(\overline{\boldsymbol{y}}^\Delta, t)$ is computed from an upper bound $\overline{\boldsymbol{y}}^\Delta$ on the logit differences $\boldsymbol{y}^\Delta := \boldsymbol{y} - y_t$ obtained via convex relaxations as described above and then plugged into the standard cross-entropy loss. As this can induce strong over-regularization if the used convex relaxations are imprecise and thereby severely reduce the standard accuracy of the resulting models, current state-of-the-art certified training methods combine these bounds with adversarial training (De Palma et al., 2022; Müller et al., 2023; Mao et al., 2023a; De Palma et al., 2024). Throughout this work we will focus on Certified Training and Robustness Guarantees only for the $\ell_\infty$-norm, i.e., $\mathcal{B}^\epsilon(\boldsymbol{x}) = \{\boldsymbol{x}' \mid ||\boldsymbol{x} - \boldsymbol{x}'||_\infty \leq \epsilon\}$, as this is the most common setting in the deterministic certified defences literature. In the following, we introduce some popular convex relaxations used for neural network verification and training.

## 2.2 Convex Relaxations

We now discuss four popular convex relaxations of different precision, investigated in this work.

**IBP** Interval bound propagation (Mirman et al., 2018; Gehr et al., 2018; Gowal et al., 2018) only considers elementwise, constant bounds of the form $\boldsymbol{l} \leq \boldsymbol{v} \leq \boldsymbol{u}$. Affine layers $\boldsymbol{y} = \boldsymbol{W}\boldsymbol{v} + \boldsymbol{b}$ are thus also relaxed as

$$\frac{\boldsymbol{W}(\boldsymbol{l}+\boldsymbol{u})-|\boldsymbol{W}|(\boldsymbol{u}-\boldsymbol{l})}{2} + \boldsymbol{b} \leq \boldsymbol{W}\boldsymbol{v} + \boldsymbol{b} \leq \frac{\boldsymbol{W}(\boldsymbol{l}+\boldsymbol{u})+|\boldsymbol{W}|(\boldsymbol{u}-\boldsymbol{l})}{2} + \boldsymbol{b}, \tag{3}$$

where $|\cdot|$ is the elementwise absolute value. ReLU functions are relaxed by their concrete lower and upper bounds $\mathrm{ReLU}(\boldsymbol{l}) \leq \mathrm{ReLU}(\boldsymbol{v}) \leq \mathrm{ReLU}(\boldsymbol{u})$, illustrated in Figure 2a.

**Hybrid Box (HBox)** The HBox relaxation is an instance of Hybrid Zonotope (Mirman et al., 2018) which combines the exact encoding of affine transformations from the DEEPZ or Zonotope domain (Singh et al., 2018; Wong & Kolter, 2018; Weng et al., 2018; Wang et al., 2018) with the simple IBP relaxation of unstable ReLUs, illustrated in Figure 2a. While less precise than DEEPZ, HBox ensures constant instead of linear representation size in the network depth, making its computation much more efficient.

**DeepPoly** DEEPPOLY, introduced by Singh et al. (2019), is mathematically identical to CROWN (Zhang et al., 2018) and based on recursively deriving linear bounds of the form

$$\boldsymbol{A}_l \boldsymbol{x} + \boldsymbol{a}_l \leq \boldsymbol{v} \leq \boldsymbol{A}_u \boldsymbol{x} + \boldsymbol{a}_u \tag{4}$$

on the outputs of every layer. While this handles affine layers exactly, ReLU layers $\boldsymbol{y} = \mathrm{ReLU}(\boldsymbol{v})$ are relaxed neuron-wise, using one of the two relaxations illustrated in Figure 2b:

$$\boldsymbol{\lambda}\boldsymbol{v} \leq \mathrm{ReLU}(\boldsymbol{v}) \leq (\boldsymbol{v} - \boldsymbol{l})\frac{\boldsymbol{u}}{\boldsymbol{u} - \boldsymbol{l}}, \tag{5}$$

where product and division are elementwise. The lower-bound slope $\boldsymbol{\lambda} = \mathbb{1}_{|\boldsymbol{u}|>|\boldsymbol{l}|}$ is chosen depending on the input bounds $l$ and $u$ to minimize the area between the upper and lower bounds in the input-output space for each neuron separately. Crucially, a minor change in the input bounds can thus lead to a large change in output bounds when using the DEEPPOLY relaxation.

**CROWN-IBP**    To reduce the computational complexity of DEEPPOLY, CROWN-IBP (Zhang et al., 2020) uses the cheaper but less precise IBP bounds to compute the concrete upper- and lower-bounds $\boldsymbol{u}$ and $\boldsymbol{l}$ on ReLU inputs required for the DEEPPOLY relaxation. To compute the final bounds on the network output DEEPPOLY is used. This reduces the computational complexity from quadratic to linear in the network depth. While CROWN-IBP is not strictly more or less precise than either IBP or DEEPPOLY, its precision empirically lies between the two (Jovanović et al., 2022).

**Relaxation Tightness**    While we rarely have strict orders in tightness (only HBOX is strictly tighter than IBP), we can empirically compare the tightness of different relaxations given a network to analyze. Jovanović et al. (2022) propose to measure the tightness of a relaxation as the AUC score of its certified accuracy over perturbation radius curve. This metric implies the following empirical tightness ordering IBP < HBOX < CROWN-IBP < DEEPPOLY (Jovanović et al., 2022), which agrees well with our intuition.

### 2.3   The Paradox of Certified Training

When training networks for robustness with convex relaxations, higher robustness is achieved by sacrificing standard accuracy. Usually, more precise relaxations induce less overapproximation and thus less regularization, potentially leading to better standard and certified accuracy. However, empirically the least precise relaxation, IBP, dominates the more precise methods, e.g., DEEPPOLY, with respect to both certified and standard accuracy (see the left-hand side of Figure 1). This is all the more surprising given that state-of-the-art certified training methods introduce artificial unsoundness into these IBP bounds to improve tightness at the cost of soundness to reduce regularisation and improve performance (Müller et al., 2023; Mao et al., 2023a; De Palma et al., 2024).

Jovanović et al. (2022) and Lee et al. (2021) explained this paradox, by showing that these more precise relaxations induce loss landscapes suffering from discontinuities, non-smoothness, and perturbation sensitivity (a proxy for difficulty to optimize with gradients), making it extraordinarily challenging for gradient-based optimization methods to find good optima. Thus the key challenge of certified training is to design a robust loss that combines tight bounds with a continuous, smooth, and insensitive loss landscape. In §3, we discuss these challenges in more detail and show how to overcome them.

## 3   Gaussian Loss Smoothing (GLS) for Certified Training

In this section, we address the optimization issues identified in §2.3—namely, discontinuity, non-smoothness, and sensitivity of the loss surface—by proposing Gaussian Loss Smoothing (GLS). We first illustrate these issues with toy examples (§3.1), highlighting how smoothing can improve loss landscapes. Then, we formalize this intuition in §3.2, showing that GLS yields continuous, smooth, and less non-convex loss surfaces. Finally, we instantiate GLS with two optimization algorithms, PGPE (§3.3) and RGS (§3.4), which enable effective training with tight (and even non-differentiable) convex relaxations.

### 3.1   Open Challenges: Discontinuity, Non-smoothness and Sensitivity

Recall §2.3, where we discussed the key challenges of certified training with tighter relaxations, namely discontinuity, non-smoothness, and sensitivity of the loss surface. We now illustrate these key challenges on a toy network and loss in Figure 3.

On the left-hand side (Original in Figure 3a), we show the DEEPPOLY lower bound of the one-neuron network $y = \text{ReLU}(x+w) + 1$ for $x \in [-1, 1]$ over the parameter $w$. As the original bound $l = 1 + \mathbb{1}_{w>0} \cdot (w-1)$ is discontinuous at $w = 0$, a gradient-based optimization method initialized at $w > 0$ will decrease $w$ until it has moved through the discontinuity and past the local minimum.

The second key factor, non-smoothness, is originally defined as the variation of loss values along the optimization trajectory. For brevity, we restrict this to the *Lipschitz continuity* of the loss function, as a Lipschitz continuous loss function has bounded variation of loss values. A function is called Lipschitz continuous if there exists a constant $L$ such that $|f(\boldsymbol{x}) - f(\boldsymbol{y})| \leq L\|\boldsymbol{x} - \boldsymbol{y}\|$ for all $\boldsymbol{x}, \boldsymbol{y}$. As DEEPPOLY has discontinuities, it is not Lipschitz continuous. We remark that Lipschitz continuity is particularly important for gradient-based optimization methods, as this controls the theoretical convergence of such methods.

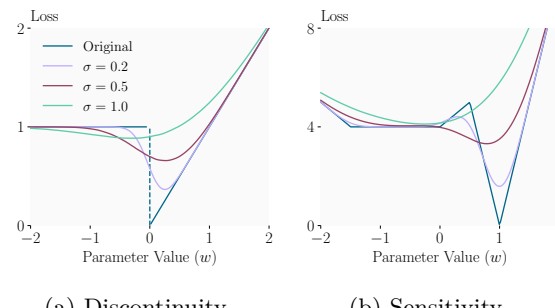

(a) Discontinuity.  (b) Sensitivity.

Figure 3: Illustrating the effect of Gaussian Loss Smoothing on the discontinuity (left) and sensitivity of loss functions (right).

The third key factor, sensitivity, can be interpreted as the difficulty to optimize with gradients. Jovanović et al. (2022) show that DEEPPOLY is more sensitive than IBP, thus gradient-based optimization methods are more likely to get stuck in bad local minima. We illustrate this with the toy function shown in Figure 3b. Here the original function has a bad local minimum for $w \in [-1.5, 0]$ that a gradient-based optimizer can get stuck in. To analyze the badness of a loss surface for gradient-based optimization, we measure the *deviation from convexity* of the loss function, defined to be $D(f) := \max_{\boldsymbol{x},\boldsymbol{y} \in \mathbb{R}^d; \lambda \in [0,1]} \delta[f; \boldsymbol{x}, \boldsymbol{y}, \lambda]$, where $\delta[f; \boldsymbol{x}, \boldsymbol{y}, \lambda] := f(\lambda \boldsymbol{x} + (1-\lambda)\boldsymbol{y}) - \lambda f(\boldsymbol{x}) - (1-\lambda)f(\boldsymbol{y})$. If a function has a non-positive deviation from convexity, it is convex, thus gradient-based methods can find global optimum. Since this directly measures non-convexity, intuitively, a function with smaller deviation from convexity is easier to optimize with gradient-based methods. We remark that sensitivity as defined in Jovanović et al. (2022) is different to the deviation from convexity, but the two are closely related in that both indicate how difficult it is to optimize a function with gradients.

## 3.2 Gaussian Loss Smoothing for Certified Training

We now discuss how Gaussian Loss Smoothing can address these challenges. The central result in this section is formalized in Theorem 3.1 (proof in §B.1):

**Theorem 3.1.** *Let the parameter $\boldsymbol{\theta} \in \mathbb{R}^d$. Let the nonnegative loss function $L(\boldsymbol{\theta}) : \mathbb{R}^d \to \mathbb{R}$ have bounded growth, that is, $L(\boldsymbol{\theta}) \exp(-\|\boldsymbol{\theta}\|^{2-\delta}) \leq M$ for some $\delta < 2$ and $M > 0$. Then, the loss smoothed by an isotropic Gaussian $\mathcal{N}(\boldsymbol{0}, \sigma^2 \boldsymbol{I})$, defined as $L_\sigma(\boldsymbol{\theta}) := \mathbb{E}_{\boldsymbol{\epsilon} \sim \mathcal{N}(\boldsymbol{0}, \sigma^2 \boldsymbol{I})} L(\boldsymbol{\theta} + \boldsymbol{\epsilon})$, is infinitely differentiable. In addition, the deviation from convexity of the smoothed loss never exceed that of the original loss, that is, $D(L_\sigma) \leq D(L)$; equality holds iff $L$ is an affine function. Further, assuming $\boldsymbol{\theta}$ is in a compact set throughout optimization, $L_\sigma$ is also Lipschitz continuous.*

Theorem 3.1 shows several desired qualities of GLS. First, it shows that GLS can turn any discontinuous loss function into a continuous one that is differentiable everywhere, as visualized in Figure 3a. Second, GLS can make the loss surface Lipschitz continuous if we optimize in a compact set, thus ensuring that the loss surface is smooth. Third, GLS can help to overcome the sensitivity issue since it provably reduces the deviation from convexity as long as the loss function is not affine. As we show in Figure 3b, depending on the standard deviation used for smoothing, the local minimum can be reduced or removed, and the loss landscape is thus more favorable. However, the choice of standard deviation is crucial. While a too-small standard deviation only has a minimal effect on loss smoothness and might not remove local minima, a too-large standard deviation can oversmooth the loss, completely removing or misaligning the minima. We again illustrate this in Figure 3b. There, a small standard deviation of $\sigma = 0.5$ works properly, while $\sigma = 0.25$ does not smooth out the local minimum, and $\sigma = 1.0$ severely misaligns the new global minimum with that of the original function. Overall, GLS has the theoretical potential to mitigate the key issues, discontinuity, non-smoothness, and sensitivity, for tight convex relaxations (as identified by Jovanović et al. (2022) and Lee et al. (2021)).

**Empirical Confirmation** To empirically confirm that GLS can mitigate discontinuity, non-smoothness, and sensitivity, we plot the original and smoothed loss landscape (along the direction of the DEEPPOLY gradient) of different relaxations for a `CNN3` and different standard deviations in Figure 4. We normalize all losses by dividing them by their value for the unperturbed weights and estimate the expectation under GLS with sampling.

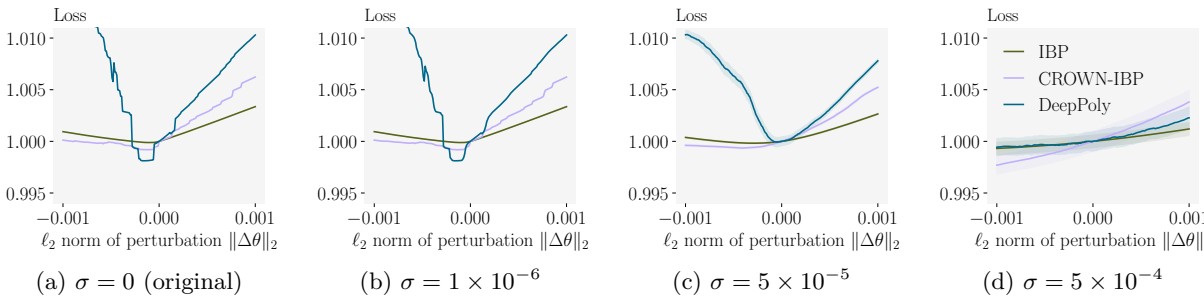

(a) $\sigma = 0$ (original)   (b) $\sigma = 1 \times 10^{-6}$   (c) $\sigma = 5 \times 10^{-5}$   (d) $\sigma = 5 \times 10^{-4}$

Figure 4: The original and Gaussian smoothed loss for different relaxations on a PGD-trained `CNN3`, evaluated along the direction of the DEEPPOLY gradient. Losses are normalized by dividing them by the values at 0, i.e., without perturbation. The smoothed loss is estimated with 128 samples and the corresponding confidence interval is shown as shaded.

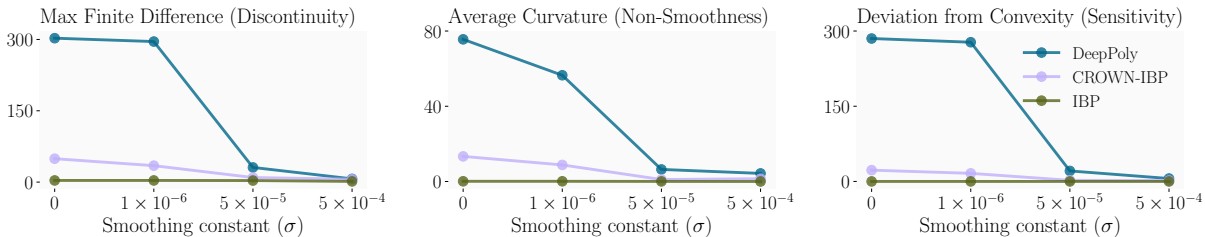

Figure 5: Attenuation of undesirable properties of the loss landscape under increasing smoothing strength. We analyze the loss landscape of the same `CNN3` as in Figure 4.

We observe that the original loss (Figure 4a) is discontinuous, non-smooth, and highly sensitive to perturbations for both CROWN-IBP and DEEPPOLY, consistent with the findings of Jovanović et al. (2022) and Lee et al. (2021). Only the imprecise IBP loss is continuous and smooth, explaining why the IBP loss is the basis for many successful certified training methods. When the loss is smoothed with small standard deviations $\sigma = 10^{-6}$ (Figure 4b), the local minimum of the DEEPPOLY loss has a slightly reduced sharpness but is still present. In addition, both the losses for DEEPPOLY and CROWN-IBP are still highly sensitive. This indicates that $\sigma$ is too small. When the standard deviation is increased to $\sigma = 5 \cdot 10^{-5}$ (Figure 4c), the undesirable local minimum of the DEEPPOLY loss is removed completely, and both losses become much smoother and less sensitive to perturbations. However, further increasing the standard deviation to $\sigma = 5 \cdot 10^{-4}$ (Figure 4d), we observe almost flat losses removing the minimum present in the underlying loss, indicating that the smoothing is too strong.

Moreover, in Figure 5, we present the evolution of three scores we used as proxies for measuring the three undesirable properties of the loss landscape as we increase the smoothing strength for the same `CNN3` network and settings used in Figure 4. Namely, we compute:

- the maximum magnitude of finite differences – defined as $\max_x \left| \frac{f(x+h)-f(x)}{h} \right|$ – as an approximation for the magnitude of the *discontinuities* in the loss function,

- the average curvature of the loss function – defined as the average magnitude of the second order derivative, estimated as $\mathbb{E}_x \left| \frac{f(x-h)-2f(x)+f(x+h)}{h^2} \right|$ – as a measure of *non-smoothness*,

- the deviation from convexity – $D(f)$ as defined in §3.1 – as a proxy for *sensitivity*,

where $f(x)$ represents the normalized loss function w.r.t. the weight perturbation as depicted in Figure 4, and $h$ is the stepsize used when sampling the loss function for making the plots, roughly equal to $1/50$ of the gradient update for a single step of gradient descent.

We observe that smoothing significantly reduces all of these scores for DEEPPOLY and CROWN-IBP, thus mitigating the optimization issues induced by the three undesirable properties.

These results empirically confirm the observations in our toy setting and predicted by our theoretical analysis, showing that GLS mitigates the issues related to the paradox of certified training.

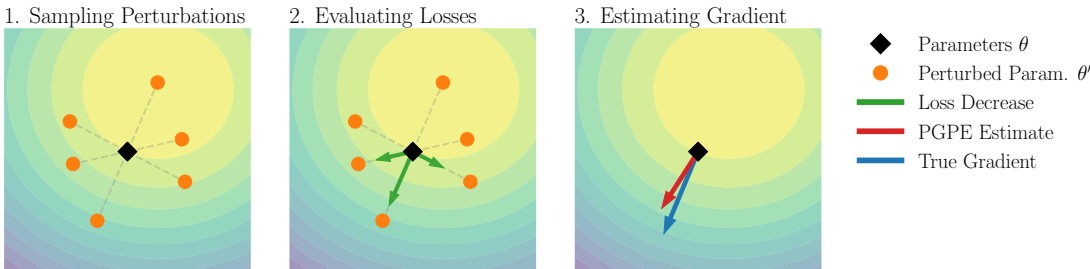

Figure 6: Illustration of PGPE. (1) We sample random perturbations $\boldsymbol{\theta}'$ (**orange** circles) symmetrically around the current parameters $\boldsymbol{\theta}$ (**black** diamond) from a Gaussian distribution $\mathcal{N}(\mathbf{0}, \boldsymbol{\sigma})$. (2) For each pair of perturbations, we evaluate the loss and compute directional differences (**green** arrows, longer arrows represent larger differences). (3) The PGPE gradient estimate (**red** arrow) is computed as a sum of all sampled directions weighted by the respective observed loss differences. The result is an approximation of the true gradient of the smoothed loss function (**blue** arrow).

Next, in §3.3 and §3.4, we show how to apply GLS using PGPE and RGS, respectively. Both PGPE and RGS being estimation-based approximations of GLS, apart from the standard deviation ($\sigma$), they also require the population size ($n_{ps}$) as a hyperparameter to estimate the gradient.

## 3.3 Policy Gradients with Parameter-based Exploration (PGPE)

PGPE (Sehnke et al., 2010) is a gradient-free optimization algorithm that optimizes the Gaussian Smoothed loss $L_\sigma(\boldsymbol{\theta}) \coloneqq \mathbb{E}_{\boldsymbol{\theta}' \sim \mathcal{N}(\boldsymbol{\theta}, \sigma^2 \boldsymbol{I})} L(\boldsymbol{\theta}')$ using a zeroth-order method, where the loss is not evaluated at a single parameterization of the network, but rather at a (normal) distribution of parameterizations.

PGPE samples $n = n_{ps}/2$ weight perturbations $\boldsymbol{\epsilon}_i \sim \mathcal{N}(\mathbf{0}, \boldsymbol{\sigma}^2)$, and evaluates the loss on $\boldsymbol{\theta} + \boldsymbol{\epsilon_i}$, and $\boldsymbol{\theta} - \boldsymbol{\epsilon_i}$, computing $r_i^+ = L(\boldsymbol{\theta} + \boldsymbol{\epsilon_i})$ and $r_i^- = L(\boldsymbol{\theta} - \boldsymbol{\epsilon_i})$. These pairs of symmetric points are then used to compute gradient estimates with respect to both the mean of the weight distribution $\boldsymbol{\theta}$ and its standard deviation $\boldsymbol{\sigma}$: $\nabla_{\boldsymbol{\theta}} \hat{L}_{\boldsymbol{\sigma}}(\boldsymbol{\theta}) \propto \sum_i \boldsymbol{\epsilon}_i (r_i^+ - r_i^-)$ and $\nabla_{\boldsymbol{\sigma}} \hat{L}_{\boldsymbol{\sigma}}(\boldsymbol{\theta}) \propto \sum_i \left( \frac{r_i^+ + r_i^-}{2} - b \right) \frac{\boldsymbol{\epsilon}_i^2 - \boldsymbol{\sigma}^2}{\boldsymbol{\sigma}}$, where $b = \frac{1}{2n} \sum_i \left( r_i^+ + r_i^- \right)$ is called baseline loss and is the average of loss values over all $2n$ samples. Figure 6 visualizes such a gradient estimate. The gradient approximations $\nabla_{\boldsymbol{\theta}} \hat{L}_{\boldsymbol{\sigma}}(\boldsymbol{\theta})$ and $\nabla_{\boldsymbol{\sigma}} \hat{L}_{\boldsymbol{\sigma}}(\boldsymbol{\theta})$ are used to update the mean weights $\boldsymbol{\theta}$ and the standard deviation $\boldsymbol{\sigma}$, respectively. By design, PGPE approximately optimizes the Gaussian smoothed loss (Sehnke et al., 2010).

As *no* backward propagation is needed to compute these gradient estimates, PGPE is comparable to neuro-evolution algorithms. In this context, it is among the best-performing methods for supervised learning (Lange et al., 2023). This property also allows us to apply it for training with tighter, but non-differentiable bounding methods, such as $\alpha$-CROWN (Xu et al., 2020).

## 3.4 Randomized Gradient Smoothing (RGS)

While the loss smoothing induced by the sampling procedure of PGPE leads to a provably continuous and infinitely differentiable loss surface, it can be costly to compute. To reduce the training costs, we propose to approximate GLS by RGS (Duchi et al., 2012). RGS approximates the gradient of the smoothed loss by sampling $n_{ps}$ perturbations $\boldsymbol{\epsilon}_i \sim \mathcal{N}(\mathbf{0}, \boldsymbol{\sigma}^2)$ and then averaging the gradients of the loss function for the perturbed network weights $\boldsymbol{\theta} + \boldsymbol{\epsilon}_i$:

$$\nabla_\theta \hat{L}_\sigma(\theta) \propto \frac{1}{n_{ps}} \sum_i \nabla_\theta L(\theta + \epsilon_i). \tag{6}$$

While RGS, which approximates in first-order does not provably recover in expectation the gradient of the smoothed loss when the original function is discontinuous (see §B.3), Duchi et al. (2012) have shown its empirical effectiveness, even with a tiny sample size ($n_{ps} = 2$). Therefore, we apply this alternative to study the performance of GLS in larger networks, as RGS requires much fewer samples than PGPE and thus scales

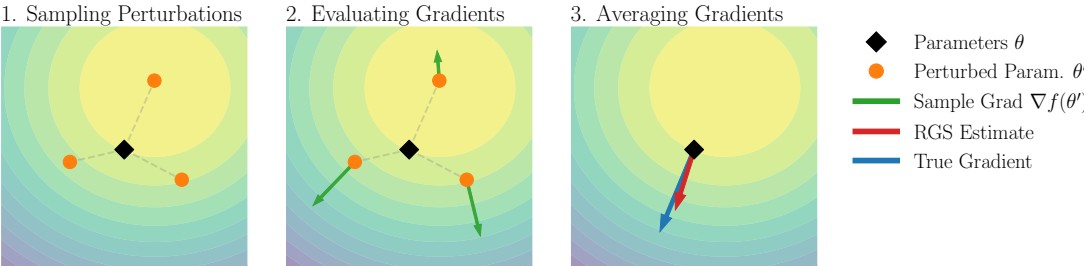

Figure 7: Illustration of RGS. (1) We sample random perturbations $\boldsymbol{\theta}'$ (**orange** circles) around the current parameters $\boldsymbol{\theta}$ (**black** diamond) from a Gaussian distribution $\mathcal{N}(\mathbf{0}, \boldsymbol{\sigma})$. (2) For each perturbation, we evaluate the gradient (**green** arrows). (3) The RGS gradient estimate (**red** arrow) is computed as the average of the sampled gradients. The result approximates the true gradient of the smoothed loss function (**blue** arrow).

better. Further, contrary to before, $\sigma$ is now a hyperparameter that needs to be tuned rather than learned. A comparison of training costs is included in §F.6, where RGS is shown to be up to 40 times faster than PGPE.

## 4 Experimental Evaluation

We now extensively evaluate the effect of GLS via PGPE and RGS on the training characteristics of different relaxation methods. First, we show in §4.2 that PGPE enables training with tight relaxations, even when the relaxation is not differentiable. Second, we demonstrate in §4.4 that RGS scales GLS training to deeper networks, surpassing the performance of the SOTA methods on the same network architecture in many settings. The impact of different hyperparameters on the performance of the proposed methods is studied in §C, and a comparison of PGPE and RGS is provided in §D.2. Overall, our results show that GLS can enable certified training with tight relaxations.

### 4.1 Experimental Setup

We implement all certified training methods in PyTorch (Paszke et al., 2019) and conduct experiments on MNIST (LeCun et al., 2010), CIFAR-10 (Krizhevsky et al., 2009) and TINYIMAGENET (Le & Yang, 2015) using $l_\infty$ perturbations and versions of the `CNN3` and `CNN5` architectures (see Table 9 in §F). For more details on the experimental setting including all hyperparameters, see §F.

**Standard Certified Training** For standard certified training using back-propagation (referred to below as GRAD for clarity), we use similar hyperparameters as in the literature and initialize all models using the IBP initialization proposed by Shi et al. (2021). In particular, we also use the Adam optimizer (Kingma & Ba, 2015), follow their learning rate and $\epsilon$-annealing schedule, use the same batch size and gradient clipping threshold, and use the same $\epsilon$ for training and certification in all settings. For the state-of-the-art methods SABR (Müller et al., 2023), STAPS (Mao et al., 2023a), and MTL-IBP (De Palma et al., 2024), we conduct an extensive optimization of their network-specific hyperparameters and only report the best results.

**PGPE Training** As PGPE training is computationally expensive, we initialize from an adversarially trained (PGD, (Madry et al., 2018)) model. This can be seen as a warm-up stage as is common also for other certified training methods (Shi et al., 2021; Müller et al., 2023; Mao et al., 2023a). We only use $\epsilon$-annealing for the larger perturbation magnitudes on both MNIST and CIFAR-10 and choose the learning rate based on stability at the beginning of the training. Unless indicated otherwise, we run the PGPE algorithm with a population size of $n_{ps} = 256$ and an initial standard deviation for weight sampling of $\sigma_{\mathrm{PGPE}} = 10^{-3}$.

**RGS Training** We train our RGS models using the same hyperparameters as for GRAD training. We use a population size of $n_{ps} = 2$ and an initial standard deviation of $\sigma_{\mathrm{RGS}} = 10^{-3}$. As RGS does not dynamically adjust the standard deviation, we choose to decay it at the same time steps as the learning rate. More details about the hyperparameters used can be found in §F.

Table 1: Comparison of the Natural, Certified and Adversarial Accuracies of `CNN3` networks trained using different convex relaxations on the MNIST and CIFAR-10 datasets for different perturbation sizes ($\epsilon_\infty$). We compare GLS-based algorithms PGPE (Sehnke et al., 2010) and RGS (Duchi et al., 2012) with the baseline approach (gradients obtained directly by backpropagation – GRAD). We use the state-of-the-art method MN-BAB (Ferrari et al., 2022) for certification.

| Dataset | $\epsilon_\infty$ | Convex Relaxation | Natural Accuracy [%] | | | Certified Accuracy [%] | | | Adversarial Accuracy [%] | | |
|---|---|---|---|---|---|---|---|---|---|---|---|
| | | | GRAD | PGPE | RGS | GRAD | PGPE | RGS | GRAD | PGPE | RGS |
| MNIST | 0.1 | IBP | 96.02 | 94.52 | 96.13 | **91.23** | 87.02 | 91.77 | 91.23 | 87.03 | 91.77 |
| | | HBOX | 94.79 | 96.12 | 95.52 | 88.18 | 90.57 | 90.13 | 88.18 | 90.58 | 90.15 |
| | | CROWN-IBP | 94.33 | 96.69 | 96.74 | 88.76 | 90.23 | 91.05 | 88.77 | 90.25 | 91.10 |
| | | DEEPPOLY | 95.95 | 97.44 | 97.37 | 90.04 | **91.53** | **91.88** | 90.08 | 91.79 | 92.03 |
| | 0.3 | IBP | 91.02 | 89.16 | 91.99 | **77.23** | 74.00 | **77.07** | 77.27 | 74.08 | 77.15 |
| | | HBOX | 83.75 | 86.58 | 83.81 | 57.86 | 70.52 | 58.37 | 57.92 | 70.66 | 58.69 |
| | | CROWN-IBP | 86.97 | 90.57 | 88.86 | 70.55 | 71.95 | 71.91 | 70.56 | 72.24 | 71.94 |
| | | DEEPPOLY | 85.70 | 91.05 | 88.51 | 66.69 | **74.28** | 71.36 | 66.70 | 74.98 | 71.49 |
| CIFAR-10 | 2/255 | IBP | 48.05 | 44.55 | 47.70 | **37.69** | 34.09 | 37.28 | 37.70 | 34.10 | 37.28 |
| | | CROWN-IBP | 44.49 | 51.19 | 53.74 | 35.75 | 37.51 | 41.00 | 35.75 | 37.65 | 41.46 |
| | | DEEPPOLY | 47.70 | 54.17 | 54.93 | 36.72 | **38.95** | **41.14** | 36.72 | 40.20 | 42.03 |
| | 8/255 | IBP | 34.63 | 30.48 | 33.23 | **25.72** | 21.75 | **24.56** | 25.74 | 21.75 | 24.58 |
| | | CROWN-IBP | 31.60 | 32.36 | 35.17 | 22.66 | 21.40 | 23.92 | 22.66 | 21.42 | 24.18 |
| | | DEEPPOLY | 33.06 | 31.37 | 35.61 | 22.97 | **22.19** | 23.81 | 22.98 | 22.19 | 24.21 |

**Certification** We use the state-of-the-art complete verification method MN-BAB (Ferrari et al., 2022) with the same settings as used by Müller et al. (2023) for all networks independently of the training method. We note that this is in contrast to Jovanović et al. (2022) who used the same relaxation for training and verification. By doing this, we aim to assess true robustness regardless of the tightness of different relaxations.

## 4.2 GLS Enables Training with Tight Relaxations

We first compare the performance of training with various differentiable convex relaxations using either standard backpropagation (GRAD) or GLS-based methods (PGPE and RGS). The result is shown in Table 1.

**GRAD Training** We train the same `CNN3` on MNIST and CIFAR-10 at the established perturbation magnitudes using standard certified training with IBP, HBOX, CROWN-IBP, and DEEPPOLY. We observe that across all these settings IBP dominates the other methods both in terms of standard and certified accuracy, confirming the paradox of certified training. Specifically, HBOX, CROWN-IBP, and DEEPPOLY tend to perform similarly, with CROWN-IBP being significantly better at MNIST $\epsilon = 0.3$, indicating that when the loss is discontinuous, non-smooth and sensitive, tightness of the training relaxation is less relevant.

**PGPE Training** Training the same `CNN3` with PGPE in the same settings we observe that the performance ranking changes significantly (see Table 1). Now, training with IBP performs strictly worse than training with DEEPPOLY across all datasets and perturbation sizes. In fact, the more precise DEEPPOLY bounds now yield the best certified accuracy across all settings, even outperforming GRAD-based training methods at low perturbation radii. Interestingly, IBP still yields better certified accuracy at large perturbation radii than HBOX and CROWN-IBP, although at significantly worse natural accuracies. This is likely because more severe regularization is required in these settings. For a more detailed discussion on the issue of certified training for large perturbations see §E.

While DEEPPOLY + PGPE outperforms DEEPPOLY + GRAD in almost all settings in Table 1 on the same network architecture, sometimes by a wide margin, it does not reach the general SOTA results of classic and heavily optimized GRAD training methods. We believe this is caused by three key factors: First, PGPE computes a gradient approximation in an $\frac{n_{\mathrm{ps}}}{2}$-dimensional subspace. To cover the full parameter space, we would need the population size $n_{\mathrm{ps}}$ to be twice the number of network parameters, which is computationally intractable even for small networks. Thus, we only get low-dimensional gradient approximations, slowing

down training (see Table 5 and Figure 9). Second, again due to the high cost of training with PGPE, we used relatively short training schedules and were unable to optimize hyperparameters for the different settings. Finally, PGPE-based certified training is less optimized, compared to standard certified training which has been extensively optimized over the past years (Shi et al., 2021; Müller et al., 2023; De Palma et al., 2024).

**RGS training** When applying RGS training to the same `CNN3` architecture, we observe that RGS significantly improves the performace of training with tighter relaxations in all settings. In particular, DEEPPOLY + RGS outperforms all other methods in the case of small perturbations, while IBP-GRAD is still the best method for large perturbations. We note that this is potentially because RGS, as a first-order approximation of GLS, does not necessarily enjoy the continuity that GLS brings. Still, the performance improvements point toward the potential of RGS to alleviate the issues of tight relaxations, while also being able to scale to deeper networks, as we show in §4.4.

**Takeaway:** GLS-based methods like PGPE and RGS enable effective training with tight convex relaxations, overcoming the paradox where looser bounds (e.g., IBP) typically outperform tighter ones. PGPE boosts certified accuracy with DeepPoly but is limited by its costly and imperfect gradient estimation, while RGS offers scalable improvements, especially for small perturbations, highlighting GLS's potential for stable and tight certified training.

### 4.3 PGPE enables non-differentiable relaxations

Next, we show that PGPE has a unique benefit in that it allows training with non-differentiable relaxations, which we demonstrate by training with the non-differentiable $\alpha$-CROWN relaxation. Since $\alpha$-CROWN is even more expensive than DEEPPOLY, we train it with a smaller version of `CNN3` called `CNN3-tiny` and set the number of iterations in $\alpha$-CROWN slope optimization to be merely 1. Table 2 shows that training with $\alpha$-CROWN-PGPE

Table 2: Accuracies of `CNN3-tiny` on MNIST $\epsilon = 0.1$ trained with different algorithms.

| Method | Nat. [%] | Cert. [%] | Adv. [%] |
|---|---|---|---|
| IBP-GRAD | 89.76 | 82.46 | 82.48 |
| DEEPPOLY-GRAD | 91.27 | 82.04 | 82.05 |
| DEEPPOLY-PGPE | 91.94 | 85.00 | 85.04 |
| $\alpha$-CROWN-PGPE | **92.15** | **85.15** | **85.17** |

further improves the certified accuracy compared to training with DEEPPOLY-PGPE. This confirms that PGPE can be used to train with non-differentiable relaxations, resulting in even better robustness-accuracy trade-offs. We remark that PGPE is not limited to $\alpha$-CROWN, but can be used with any non-differentiable relaxation, including those relying on branch and bound-based procedures or multi-neuron constraints. Although these methods are computationally expensive and thus may be only applied in training small networks, they are particularly useful in safety-critical applications such as aircraft control (Owen et al., 2019) or embedded medical devices (Shoeb et al., 2009), where models are usually even smaller.

### 4.4 RGS Scales GLS Training

We have demonstrated the empirical advantages of GLS instantiated with PGPE. However, as PGPE is computationally expensive and limited to small models, more scalable methods are required to train larger networks. In this section, we extensively evaluate RGS, showing that its efficiency allows us to scale to larger models, surpassing the performance of the SOTA methods on the same network architecture on standard evaluation settings when $\epsilon_\infty$ is relatively small.

RGS overcomes the low-rank gradient and computational cost issues of PGPE: even with a small population size (hence low training costs), we obtain full-rank gradient approximations, enabling faster and better optimization and allowing us to even scale our experiments to TINYIMAGENET. We analyze the results of training with RGS on the `CNN5` and `CNN5-L` (a wider version of `CNN5`) architectures and compare them with IBP and the SOTA GRAD-based methods (Mao et al., 2024) trained on `CNN7` in Table 3. Encouragingly, RGS significantly boosts the performance of DEEPPOLY training. We observe that DEEPPOLY + RGS dominates all other methods, substantially improving even over state-of-the-art GRAD-based methods with hyperparameters fine-tuned on `CNN5` and `CNN5-L`. Further, the performance of DEEPPOLY + RGS on the small `CNN5` becomes comparable to the performance of GRAD-IBP on the much larger `CNN7` architecture used by recent SOTA methods, and the `CNN5-L` trained with DEEPPOLY-RGS exceeds the performance of `CNN7` trained with IBP by a large margin. These results agree well with our expectation that bound tightness becomes

Table 3: Comparison between networks trained with DEEPPOLY-RGS, CROWN-IBP-RGS and SOTA GRAD methods on small perturbation settings. The best performance for each dataset and architecture is **highlighted**. Numbers in *italic* represent results for GRAD methods obtained on the SOTA `CNN7` architecture, which is more than 10 times larger than the `CNN5` and `CNN5-L` architectures.

| Dataset | Network (params.) | Method | Nat. Acc. [%] | Cert. Acc. [%] | Adv. Acc. [%] |
|---|---|---|---|---|---|
| MNIST $\epsilon_\infty = 0.1$ | CNN5 (166K) | IBP | 97.94 | 95.82 | 95.83 |
| | | SABR | 98.81 | 96.28 | 96.31 |
| | | STAPS | 98.74 | 96.05 | 96.09 |
| | | MTL-IBP | 98.74 | 96.25 | 96.29 |
| | | CROWN-IBP | 98.19 | 95.42 | 95.42 |
| | | CROWN-IBP-RGS | 98.43 | 95.64 | 95.65 |
| | | DEEPPOLY | 98.50 | 95.95 | 95.97 |
| | | DEEPPOLY-RGS | **98.97** | **97.15** | **97.16** |
| | CNN5-L (1.25M) | MTL-IBP | 98.91 | 97.17 | 97.33 |
| | | DEEPPOLY-RGS | **99.21** | **97.61** | **97.76** |
| | CNN7 (13.3M) | IBP | *98.87* | *98.26* | *98.27* |
| | | TAPS | *99.16* | *98.52* | *98.58* |
| CIFAR-10 $\epsilon_\infty = 2/255$ | CNN5 (281K) | IBP | 54.92 | 45.36 | 45.36 |
| | | SABR | 66.73 | 52.11 | 52.55 |
| | | MTL-IBP | 67.03 | 53.81 | 55.18 |
| | | CROWN-IBP | 60.91 | 49.45 | 49.68 |
| | | CROWN-IBP-RGS | 63.22 | 50.73 | 51.18 |
| | | DEEPPOLY | 65.43 | 53.16 | 54.10 |
| | | DEEPPOLY-RGS | **67.88** | **54.91** | **56.12** |
| | CNN5-L (1.25M) | MTL-IBP | 70.60 | 56.36 | 59.05 |
| | | DEEPPOLY-RGS | **72.64** | **59.34** | **61.23** |
| | CNN7 (17.2M) | IBP | *67.49* | *55.99* | *56.10* |
| | | MTL-IBP | *78.82* | *64.41* | *67.69* |
| TINYIMAGENET $\epsilon_\infty = 1/255$ | CNN5 (1.17M) | IBP | 19.55 | 13.92 | 13.93 |
| | | MTL-IBP | 26.92 | 18.07 | 18.16 |
| | | CROWN-IBP-LF | 21.91 | 16.43 | 16.43 |
| | | CROWN-IBP-LF-RGS | 22.97 | 16.89 | 16.89 |
| | | DEEPPOLY-RGS | **27.84** | **19.73** | **20.40** |
| | CNN7 (17.3M) | IBP | *26.77* | *19.82* | *19.84* |
| | | MTL-IBP | *35.97* | *27.73* | *28.49* |

increasingly important with network depth, as overapproximation errors can grow exponentially with depth (Shi et al., 2021; Müller et al., 2023; Mao et al., 2023b). We remark that scaling DEEPPOLY-RGS to `CNN7` used by the SOTA methods is still infeasible due to the high computational cost of evaluating DEEPPOLY (RGS only doubles the cost!), but we show that RGS can still be used with the cheaper CROWN-IBP relaxation on this architecture in Table 6 in §D.1.

### 4.5 RGS Training in Large Perturbation Settings

In Table 4 we provide experimental data for training `CNN5` networks using DEEPPOLY + RGS. We observe that while DEEPPOLY + RGS manages to obtain similar natural accuracies with gradient-based IBP, the certified accuracies are significantly lower. This is likely because to gain certifiability for the large epsilon settings the networks require a stronger regularisation than the DEEPPOLY relaxation can provide.

This is in agreement with the findings of Mao et al. (2024), which after extensive hyperparameter tuning, show that IBP trained networks can obtain very close performance to SOTA methods in the large perturbation settings. For example, while the SOTA method, MTL-IBP, improves IBP by more than 10% for CIFAR-10, $\epsilon = 2/255$, it merely improves IBP by 0.13% for CIFAR-10, $\epsilon = 8/255$, representing a roughly 100x reduction in relative improvement. We observe a similar pattern in our experiments with standard certified training methods on `CNN5`. A more detailed discussion can be found in §E.

Table 4: Accuracies of a `CNN5` depending on training method.

| Dataset | Method | Nat. [%] | Cert. [%] | Adv. [%] |
|---|---|---|---|---|
| MNIST $\epsilon_\infty = 0.3$ | IBP (used as init) | 94.95 | 87.71 | 87.80 |
| | SABR | **97.78** | 88.26 | **89.33** |
| | MTL-IBP | 97.08 | 88.68 | 88.95 |
| | DEEPPOLY-RGS | 95.79 | 87.04 | 87.17 |
| | DEEPPOLY-RGS (IBP) | 95.47 | **88.69** | 88.79 |
| CIFAR-10 $\epsilon_\infty = 8/255$ | IBP (used as init) | 41.05 | 29.12 | 29.14 |
| | SABR | 43.30 | 29.50 | 29.55 |
| | MTL-IBP | **44.53** | **29.62** | **29.73** |
| | DEEPPOLY-RGS | 40.10 | 25.25 | 25.93 |
| | DEEPPOLY-RGS (IBP) | 41.66 | 29.25 | 29.31 |

To verify that training with tighter relaxations can still yield improvements in the large perturbation settings, we initialize the `CNN5` networks with IBP-trained weights and further train them with DEEPPOLY + RGS. The results are shown in Table 4, denoted by DEEPPOLY-RGS (IBP). We observe that training with DEEPPOLY + RGS increases both natural and certified accuracies when compared to the IBP-trained initialization, with certified accuracy reaching a similar level with MTL-IBP on MNIST 0.3. However, on CIFAR-10 $\epsilon = 8/255$, this is still weaker than the SOTA MTL-IBP, although it strictly improves over IBP. This demonstrates the ability of tighter relaxations to still improve training in the large perturbation settings, but more work is needed to surpass the performance of SOTA methods.

## 5 Discussion and Limitations

This work shows the promise of Gaussian Loss Smoothing (GLS) to enable certified training with tight relaxations. PGPE and RGS, our proposed methods implementing GLS, achieve strong performance empirically. However, there are several limitations and challenges that need to be addressed in future work. First, GLS provably mitigates the discontinuity, non-smoothness, and perturbation sensitivity issues identified, but it is unknown whether these are all the factors contributing to the paradox of certified training. Future work should investigate other potential factors and how they can be addressed. Second, while our methods achieve strong performance, they are computationally expensive. Future work should focus on more computationally efficient smoothing approaches. Finally, we present a first step towards training with tight relaxations, but our methods could be further optimized, similar to how IBP-based methods have been optimized over the years. Overall, our work opens up a new direction for certified training using tight relaxations, and we hope it will inspire future work in this area.

## 6 Related Work

To guarantee robustness of neural networks, two main approaches have been proposed. Randomized smoothing (Cohen et al., 2019; Salman et al., 2019; 2020; Jeong et al., 2021; Horváth et al., 2022; Vaishnavi et al., 2022; Jeong et al., 2023; Sun et al., 2024) applies random perturbation on the input and draw statistical robustness guarantees on the smoothed classifier. To provide deterministic guarantees, convex relaxations of the neural network are commonly adopted (Mirman et al., 2018; Wong et al., 2018; Singh et al., 2018; Zhang et al., 2018; Singh et al., 2019). These methods are limited in completeness (Mirman et al., 2022; Baader et al., 2024; Mao et al., 2025), thus the most effective certification algorithms combine branch-and-bound with convex relaxations to yield complete certification (Bunel et al., 2020; Xu et al., 2021; Wang et al., 2021; Ferrari et al., 2022). However, these methods are computationally expensive and do not scale well to large networks. Thus, neural networks tailored for certification have been widely studied (Gowal et al., 2018; Balunović & Vechev, 2020; Zhang et al., 2020; Shi et al., 2021; De Palma et al., 2022; Müller et al., 2023; Mao et al., 2023a;b; De Palma et al., 2024). Surprising phenomena, however, have been observed when training certified models, where the performance of certified models trained with more accurate bounds may not exceed that

of models trained with less accurate bounds. While some reasons have been identified (Jovanović et al., 2022; Lee et al., 2021), it is not yet clear how to solve this issue. This work examines whether solving the identified optimization difficulties can help solve this problem.

## 7 Conclusion

This work shows that the three issues contributing to the paradox of certified training identified by prior works, namely discontinuity, non-smoothness, and perturbation sensitivity, can be mitigated by Gaussian Loss Smoothing (GLS), based on sound theoretical analyses. We instantiate GLS with two methods: Policy Gradients with Parameter-based Exploration (PGPE) and Randomized Gradient Smoothing (RGS). Empirically, we demonstrate that both improve training with tight relaxations, presenting a solid step towards overcoming the paradox. Further, we show that both methods have unique advantages: PGPE allows training with non-differentiable relaxations, while RGS scales better. Our results confirm the importance of loss continuity, smoothness, and insensitivity in certified training, and pave the way for future work to leverage tighter relaxations for certified training.

## Reproducibility Statement

We release the complete code used for our experiments at github.com/stefanrzv2000/GLS-Cert-Training. A detailed description of the experimental setup and hyperparameters is provided in §F.

## Acknowledgements

This research was partially funded by the Ministry of Education and Science of Bulgaria (support for INSAIT, part of the Bulgarian National Roadmap for Research Infrastructure).

This work has been done as part of the EU grant ELSA (European Lighthouse on Secure and Safe AI, grant agreement no. 101070617) and the SERI grant SAFEAI (Certified Safe, Fair and Robust Artificial Intelligence, contract no. MB22.00088). Views and opinions expressed are however those of the authors only and do not necessarily reflect those of the European Union or European Commission. Neither the European Union nor the European Commission can be held responsible for them.

The work has received funding from the Swiss State Secretariat for Education, Research and Innovation (SERI).

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

## A   Notation

We use the following notation throughout this work:

- $\boldsymbol{x} \in \mathcal{X} \subseteq \mathbb{R}^d$: Input vector to the neural network, assumed to lie in an input domain $\mathcal{X}$.

- $\mathcal{Y} = \{1, \ldots, n\}$: Set of class labels for an $n$-class classification task.

- $\boldsymbol{f_\theta}(\boldsymbol{x}) \in \mathbb{R}^n$: Neural network output (logits) for input $\boldsymbol{x}$, parameterized by weights $\boldsymbol{\theta}$.

- $F(\boldsymbol{x}) = \arg\max_i \boldsymbol{f_\theta}(\boldsymbol{x})_i$: Induced classifier assigning input $\boldsymbol{x}$ to the class with the highest logit.

- $\mathcal{B}_p^\epsilon(\boldsymbol{x}) = \{\boldsymbol{x}' \in \mathcal{X} \mid \|\boldsymbol{x} - \boldsymbol{x}'\|_p \leq \epsilon\}$: Closed $\ell_p$-ball of radius $\epsilon$ centered at $\boldsymbol{x}$, representing the set of admissible perturbations.

- $\mathcal{L}_{\text{CE}}(\boldsymbol{y}, t) = \ln\left(1 + \sum_{i \neq t} \exp(y_i - y_t)\right)$: Cross-entropy loss expressed in terms of the logit differences.

- $\boldsymbol{l}, \boldsymbol{u}$: Lower and upper bounds on intermediate or output activations, typically obtained via convex relaxations.

- $\boldsymbol{A}_l, \boldsymbol{b}_l$ and $\boldsymbol{A}_u, \boldsymbol{b}_u$: Coefficients defining affine lower and upper bounds on $\boldsymbol{f_\theta}(\boldsymbol{x})$ over the perturbation region.

- $\boldsymbol{y}^\Delta := \boldsymbol{y} - y_t$: Logit difference vector between logits $\boldsymbol{y}$ and the target class $y_t$.

- $\overline{\boldsymbol{y}}^\Delta$: Upper bound on the logit differences, used to compute the robust loss in certified training.

- $\boldsymbol{\theta} \in \mathbb{R}^d$: Parameter vector of the model.

- $L(\boldsymbol{\theta}) : \mathbb{R}^d \to \mathbb{R}$: Original (possibly non-smooth or discontinuous) loss function.

- $\mathcal{N}(\boldsymbol{0}, \sigma^2 \boldsymbol{I})$ or $\mathcal{N}(\boldsymbol{0}, \sigma^2)$: Isotropic Gaussian distribution with zero mean and standard deviation $\sigma$ (scalar).

- $\mathcal{N}(\boldsymbol{0}, \boldsymbol{\sigma}^2 \boldsymbol{I})$: Anisotropic Gaussian distribution with zero mean and standard deviation $\boldsymbol{\sigma}$ (vector).

- $\boldsymbol{\epsilon} \sim \mathcal{N}(\boldsymbol{0}, \sigma^2 \boldsymbol{I})$: Gaussian noise vector used for smoothing.

- $L_\sigma(\boldsymbol{\theta}) := \mathbb{E}_{\boldsymbol{\epsilon} \sim \mathcal{N}(\boldsymbol{0}, \sigma^2 \boldsymbol{I})} L(\boldsymbol{\theta} + \boldsymbol{\epsilon})$: Gaussian smoothed version of the loss.

- $D(f)$: Deviation from convexity of function $f$, defined as
  $D(f) := \max_{\boldsymbol{x}, \boldsymbol{y} \in \mathbb{R}^d, \lambda \in [0,1]} [f(\lambda \boldsymbol{x} + (1 - \lambda)\boldsymbol{y}) - \lambda f(\boldsymbol{x}) - (1 - \lambda)f(\boldsymbol{y})]$.

- $L$-Lipschitz continuity: A function $f$ is Lipschitz continuous if $\exists L > 0$ s.t. $|f(\boldsymbol{x}) - f(\boldsymbol{y})| \leq L\|\boldsymbol{x} - \boldsymbol{y}\|$ for all $\boldsymbol{x}, \boldsymbol{y}$.

- $\text{ReLU}(\cdot)$: Rectified Linear Unit activation function.

- $\mathbb{1}_{w > 0}$: Indicator function that is 1 if $w > 0$ and 0 otherwise.

- $\frac{f(x+h) - f(x)}{h}$: Finite difference approximation used to estimate discontinuity magnitude.

- $\frac{f(x-h) - 2f(x) + f(x+h)}{h^2}$: Second-order finite difference used to estimate curvature (non-smoothness).

- $\sigma, \boldsymbol{\sigma}_{PGPE}, \sigma_{RGS}$: Standard deviation of Gaussian used in GLS. A vector in the case of PGPE, a scalar hyperparameter for RGS.

- $n_{ps}$: Population size used in sampling-based GLS estimators (e.g., PGPE, RGS).

Unless otherwise noted, we focus on robustness with respect to the $\ell_\infty$ norm, i.e., $\mathcal{B}^\epsilon(\boldsymbol{x}) = \{\boldsymbol{x}' \in \mathcal{X} \mid \|\boldsymbol{x} - \boldsymbol{x}'\|_\infty \leq \epsilon\}$.

# B    Theoretical Power of GLS

## B.1    Proofs

Throughout our proof, we will denote the probability density function of the Gaussian distribution $\mathcal{N}(\mathbf{0}, \sigma^2 \mathbf{I})$ as $p_\sigma(x)$. A ball of radius $r$ is defined as $B(r) := \{\boldsymbol{x} \mid \|\boldsymbol{x}\| \leq r\}$. Without explicit mention, the norms are $L_2$ norm.

**Lemma B.1.** *Assume the existence of the smoothed loss function. Gaussian Loss Smoothing is equivalent to performing a convolution of the loss function with a Gaussian kernel, that is, $L_\sigma(\boldsymbol{\theta}) = \left[L * \mathcal{N}(\mathbf{0}, \sigma^2 \mathbf{I})\right](\boldsymbol{\theta})$.*

*Proof.* Remember that $L_\sigma(\boldsymbol{\theta}) := \mathbb{E}_{\boldsymbol{\epsilon} \sim \mathcal{N}(\mathbf{0}, \sigma^2 \mathbf{I})} L(\boldsymbol{\theta} + \boldsymbol{\epsilon})$. Let $\boldsymbol{x} := \boldsymbol{\theta} + \boldsymbol{\epsilon}$ and use $p_\sigma(\boldsymbol{\epsilon}) = p_\sigma(-\boldsymbol{\epsilon})$ due to symmetry, we have

$$
\begin{aligned}
L_\sigma(\boldsymbol{\theta}) &= \mathbb{E}_{\boldsymbol{\epsilon} \sim \mathcal{N}(\mathbf{0}, \sigma^2 \mathbf{I})} L(\boldsymbol{\theta} + \boldsymbol{\epsilon}) \\
&= \int_{\mathbb{R}^d} L(\boldsymbol{\theta} + \boldsymbol{\epsilon}) p_\sigma(\boldsymbol{\epsilon}) d\boldsymbol{\epsilon} \\
&= \int_{\mathbb{R}^d} L(\boldsymbol{\theta} + \boldsymbol{\epsilon}) p_\sigma(-\boldsymbol{\epsilon}) d\boldsymbol{\epsilon} \\
&= \int_{\mathbb{R}^d} p_\sigma(\boldsymbol{\theta} - \boldsymbol{x}) L(\boldsymbol{x}) d\boldsymbol{x} \\
&= \left[L * \mathcal{N}(\mathbf{0}, \sigma^2 \mathbf{I})\right](\boldsymbol{\theta}).
\end{aligned}
\tag{7}
$$

$\square$

**Proposition B.2.** *Assume the nonnegative loss function $L(\boldsymbol{\theta}) : \mathbb{R}^d \to \mathbb{R}$ have bounded growth, that is, $L(\boldsymbol{\theta}) \exp(-\|\boldsymbol{\theta}\|^{2-\delta}) \leq M$ for some $\delta < 2$ and $M > 0$. Then, $L_\sigma$ exists and is infinitely differentiable.*

*Proof.* We prove existence of $L_\sigma$ first. Equation (7) shows that this is equivalent to prove the convergence of the integral. Given a fixed $\boldsymbol{\theta}$, $p_\sigma(\boldsymbol{\theta} - \boldsymbol{x}) \propto \exp(-\frac{1}{2\sigma^2}\|\boldsymbol{x} - \boldsymbol{\theta}\|^2)$, thus $\exists \alpha_1, \beta_1, M_1 > 0$, such that $p_\sigma(\boldsymbol{\theta} - \boldsymbol{x}) \leq \alpha_1 \exp(-\beta_1 \|\boldsymbol{x}\|^2)$ when $\|\boldsymbol{x}\| > M_1$. Therefore,

$$
\begin{aligned}
&\int_{\mathbb{R}^d \setminus B(M_1)} p_\sigma(\boldsymbol{\theta} - \boldsymbol{x}) L(\boldsymbol{x}) d\boldsymbol{x} \\
\leq{}& \int_{\mathbb{R}^d \setminus B(M_1)} \alpha_1 \exp(-\beta_1 \|\boldsymbol{x}\|^2) L(\boldsymbol{x}) d\boldsymbol{x} \\
\leq{}& \int_{\mathbb{R}^d \setminus B(M_1)} \alpha_1 \exp(-\beta_1 \|\boldsymbol{x}\|^2) M \exp(\|\boldsymbol{x}\|^{2-\delta}) d\boldsymbol{x} \\
\leq{}& \alpha_1 M \int_{\mathbb{R}^d \setminus B(M_1)} \exp(-\beta_1 \|\boldsymbol{x}\|^2 + \|\boldsymbol{x}\|^{2-\delta}) d\boldsymbol{x}.
\end{aligned}
$$

Further, $\exists \beta_2 > 0, M_2 \geq M_1$, such that $\exp(-\beta_1 \|\boldsymbol{x}\|^2 + \|\boldsymbol{x}\|^{2-\delta}) \leq \exp(-\beta_2 \|\boldsymbol{x}\|^{\delta/2})$ when $\|\boldsymbol{x}\| > M_2$. Therefore,

$$
\begin{aligned}
&\int_{\mathbb{R}^d \setminus B(M_2)} p_\sigma(\boldsymbol{\theta} - \boldsymbol{x}) L(\boldsymbol{x}) d\boldsymbol{x} \\
\leq{}& \alpha_1 M \int_{\mathbb{R}^d \setminus B(M_2)} \exp(-\beta_1 \|\boldsymbol{x}\|^2 + \|\boldsymbol{x}\|^{2-\delta}) d\boldsymbol{x} \\
\leq{}& \alpha_1 M \left[\int_{\mathbb{R}^d \setminus B(M_2)} \exp(-\beta_2 \|\boldsymbol{x}\|^{\delta/2}) d\boldsymbol{x}\right].
\end{aligned}
$$

Note that $\exp(-\beta_2\|\boldsymbol{x}\|^{\delta/2})$ decays faster than $\frac{1}{\|\boldsymbol{x}\|^2}$ and $\int_{\mathbb{R}^d\setminus B(M_2)}\frac{1}{\|\boldsymbol{x}\|^2}d\boldsymbol{x}$ is bounded. Thus, $\forall\epsilon>0$, $\exists M_3\geq M_2$, such that $\int_{\mathbb{R}^d\setminus B(M_3)}p_\sigma(\boldsymbol{\theta}-\boldsymbol{x})L(\boldsymbol{x})d\boldsymbol{x}<\epsilon$. Therefore, $L_\sigma$ exists.

Now we turn to its derivative. Using Lemma B.1 and $(f*g)'(t)=(f*g')(t)$, we know that any $n$-th (partial) derivative of $L_\sigma$ is $L*\frac{\partial^{(n)}p_\sigma}{\partial^{(n)}\boldsymbol{x}}$, where $\partial^{(n)}\boldsymbol{x}$ is a shorthand for the related variables. Since $n$-th partial derivative of a Gaussian pdf is a polynomial (Hermite polynomials) times a Gaussian pdf, we can bound it similarly to what we have done before, as we can still bound $\frac{\partial^{(n)}p_\sigma}{\partial^{(n)}\boldsymbol{x}}$ with $\alpha_1\exp(-\beta_1\|\boldsymbol{x}\|^2)$ under appropriate $\alpha_1,\beta_1,M_1$. Therefore, $L_\sigma$ is infinitely differentiable. $\square$

**Lemma B.3.** *If $f$ is continuously differentiable, then $f$ is Lipschitz continuous within a compact set.*

*Proof.* Since $f$ has continuous first-order derivative, it suffices to show that the first-order gradients are bounded. This is trivial as a continuous function is bounded within any compact set. $\square$

**Proposition B.4.** *Assume $f*g$ exists, where $g$ is a probability density function. Then, the deviation from convexity of $f*g$ is smaller than or equal to the deviation from convexity of $f$, that is, $D(f*g)\leq D(f)$. Equality holds iff $f$ is an affine function.*

*Proof.*

$$\begin{aligned}
\delta[f*g;\boldsymbol{x},\boldsymbol{y},\lambda] &= f*g(\lambda\boldsymbol{x}+(1-\lambda)\boldsymbol{y})-\lambda f*g(\boldsymbol{x})-(1-\lambda)f*g(\boldsymbol{y})\\
&= \int_{\mathbb{R}^d}\left[f(\lambda\boldsymbol{x}+(1-\lambda)\boldsymbol{y}-\boldsymbol{z})-\lambda f(\boldsymbol{x}-\boldsymbol{z})-(1-\lambda)f(\boldsymbol{y}-\boldsymbol{z})\right]g(\boldsymbol{z})d\boldsymbol{z}\\
&= \int_{\mathbb{R}^d}\delta[f;\boldsymbol{x}-z,\boldsymbol{y}-z,\lambda]g(\boldsymbol{z})d\boldsymbol{z}\\
&\leq \max_{\boldsymbol{x},\boldsymbol{y}\in\mathbb{R}^d;\lambda\in[0,1]}\delta[f;\boldsymbol{x},\boldsymbol{y},\lambda]\int_{\mathbb{R}^d}g(\boldsymbol{z})d\boldsymbol{z}\\
&= D(f)\int_{\mathbb{R}^d}g(\boldsymbol{z})d\boldsymbol{z}\\
&= D(f),
\end{aligned}$$

where we used the fact that $g$ is a probability density function, thus $\int_{\mathbb{R}^d}g(\boldsymbol{z})d\boldsymbol{z}=1$. The above equality holds iff $\delta[f;\boldsymbol{x},\boldsymbol{y},\lambda]$ is a constant function. Therefore, $D(f*g)=\max_{\boldsymbol{x},\boldsymbol{y}\in\mathbb{R}^d;\lambda\in[0,1]}\delta[f*g;\boldsymbol{x},\boldsymbol{y},\lambda]\leq D(f)$. Note that to take equality, it is necessary that $\delta[f*g;\boldsymbol{x},\boldsymbol{y},\lambda]=D(f)$ for some $\boldsymbol{x},\boldsymbol{y},\lambda$, thus $\delta[f;\boldsymbol{x},\boldsymbol{y},\lambda]$ still has to be a constant function. On the other hand, if $\delta[f;\boldsymbol{x},\boldsymbol{y},\lambda]$ is a constant function, then $\delta[f*g;\boldsymbol{x},\boldsymbol{y},\lambda]=D(f)$ for all $\boldsymbol{x},\boldsymbol{y},\lambda$, thus $D(f*g)=D(f)$. Therefore, $D(f*g)=D(f)$ iff $\delta[f;\boldsymbol{x},\boldsymbol{y},\lambda]$ is a constant function.

Now we show that $\delta[f;\boldsymbol{x},\boldsymbol{y},\lambda]$ is a constant function iff $f$ is an affine function. If $f$ is an affine function, then $\delta[f;\boldsymbol{x},\boldsymbol{y},\lambda]=f(\lambda\boldsymbol{x}+(1-\lambda)\boldsymbol{y})-\lambda f(\boldsymbol{x})-(1-\lambda)f(\boldsymbol{y})=0$, thus $\delta[f;\boldsymbol{x},\boldsymbol{y},\lambda]$ is a constant function. On the other hand, if $\delta[f;\boldsymbol{x},\boldsymbol{y},\lambda]$ is a constant function, then $\exists C$ such that $\delta[f;\boldsymbol{x},\boldsymbol{y},\lambda]=C$ for all $\boldsymbol{x},\boldsymbol{y},\lambda$. Let $\boldsymbol{x}=\boldsymbol{y}=\boldsymbol{0}$, then $C=f(\boldsymbol{0})-f(\boldsymbol{0})=0$, thus $f(\lambda\boldsymbol{x}+(1-\lambda)\boldsymbol{y})=\lambda f(\boldsymbol{x})+(1-\lambda)f(\boldsymbol{y})$ for all $\boldsymbol{x},\boldsymbol{y},\lambda$. This means $f$ is an affine function. $\square$

**Theorem 3.1.** *Let the parameter $\boldsymbol{\theta}\in\mathbb{R}^d$. Let the nonnegative loss function $L(\boldsymbol{\theta}):\mathbb{R}^d\to\mathbb{R}$ have bounded growth, that is, $L(\boldsymbol{\theta})\exp(-\|\boldsymbol{\theta}\|^{2-\delta})\leq M$ for some $\delta<2$ and $M>0$. Then, the loss smoothed by an isotropic Gaussian $\mathcal{N}(\boldsymbol{0},\sigma^2\boldsymbol{I})$, defined as $L_\sigma(\boldsymbol{\theta}):=\mathbb{E}_{\boldsymbol{\epsilon}\sim\mathcal{N}(\boldsymbol{0},\sigma^2\boldsymbol{I})}L(\boldsymbol{\theta}+\boldsymbol{\epsilon})$, is infinitely differentiable. In addition, the deviation from convexity of the smoothed loss never exceed that of the original loss, that is, $D(L_\sigma)\leq D(L)$; equality holds iff $L$ is an affine function. Further, assuming $\boldsymbol{\theta}$ is in a compact set throughout optimization, $L_\sigma$ is also Lipschitz continuous.*

*Proof.* By Proposition B.2, $L_\sigma$ exists and is infinitely differentiable. Further, by Lemma B.1 and Proposition B.4, $D(L_\sigma)\leq D(L)$; equality holds iff $L$ is an affine function. Assuming $\theta$ is in a compact set, by Lemma B.3, $L_\sigma$ is Lipschitz continuous. $\square$

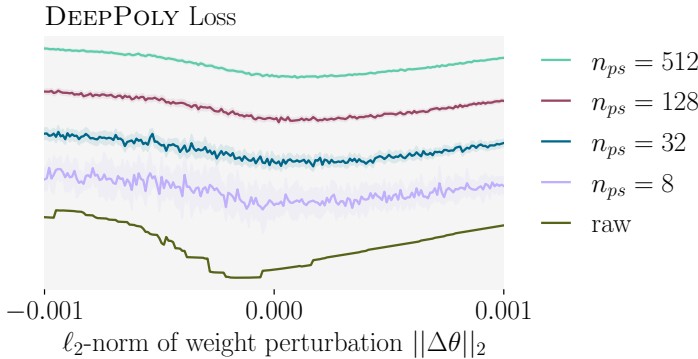

Figure 8: Effect of the population size $n_{ps}$ on the smoothness of the induced loss surface in PGPE. Note that the 5 plots have been spaced by artificially adding offsets on the y-axis. This should not be regarded as a quantitative plot ordering the magnitude of the loss, but rather as a qualitative comparison of the smoothness induced by sampling with different population sizes.

## B.2   Alignment of Local and Global Minima under Gaussian Loss Smoothing

Without loss of generality, we consider a quantized function $f(x) = \sum_{i=0}^{n} a_i I(x \in [b_i, b_{i+1}])$, where $I$ is the threshold function and $-\infty = b_0 \leq b_1 \leq \cdots \leq b_n \leq b_{n+1} = +\infty$. The global minimum of this function is $\min_i a_i$, achieved by $x \in [b_{i^*}, b_{i^*+1}]$ where $i^* \in \arg\min_i a_i$. Now, the derivative of its Gaussian smoothed loss is $f'_\sigma(x) = \frac{1}{\sigma} \sum_{i=1}^{n} (a_i - a_{i-1}) p(\frac{b_i - x}{\sigma})$, where $p$ is the p.d.f. of the standard normal distribution. One may immediately find that the minimum of the smoothed loss is scale-invariant: the minimum of $f_{c\sigma}(cx)$ with $b_i$ scaled by $c$ is the same as the minimum of $f_\sigma(x)$. Therefore, if we increase $\sigma$ to smoothen a fixed function, shallower minima with smaller widths will be smoothed out one by one. Taking $\sigma$ to $\infty$, we find that the derivative converges to zero, making the smoothed loss a constant function.

We use the same quantized function to study the effect smoothing has on the alignment of minimum points. As observed before, when we take $\sigma$ to $\infty$, the derivative on the whole domain converges to zero, so every point becomes a minimum, therefore we fail to get a proper alignment. On the other hand, by taking $\sigma$ to zero, the factor $p(\frac{b_i - x}{\sigma})$ becomes a Dirac delta function $\delta(x = b_i)$, thus every point except the boundary points becomes a local minimum, and we get the alignment of global minima. Based on these intuitions, one can pick a $\sigma$ such that narrow local minima get smoothed out, and wide local minima are left close to their original locations, thus the optimization process can be guided towards the global minimum.

## B.3   Properties of Randomized Gradient Smoothing

**Discontinuity**   Considering again the quantized function defined in §B.2, we observe that the derivative of the original function is zero almost everywhere, so the smoothed gradient estimated by RGS will also be zero. This means that RGS may fail to find the minimum of certain discontinuous functions in general. However, in practice we rarely work with quantized loss functions we used for the analysis; instead, we can model the discontinuous loss function as $h(x) = f(x) + g(x)$, where $f(x)$ is discontinuous like the quantized function and $g(x)$ is continuous. In this case, the derivative of $h$ is equal to the derivative of $g$ almost everywhere, and thus the RGS algorithm will converge to the same locations when optimizing $h$ as when optimizing $g$. If the minima of $g$ and $h$ are sufficiently aligned, we can expect RGS to find a good minimum of $h$.

**Higher Dimensions**   In higher dimensions, however, the behavior of RGS becomes unpredictable, as not every discontinuous function $h$ can be decomposed into a continuous function $g$ and a quantized function $f$ (e.g. $h(x_1, x_2) = x_1 \cdot \text{sign}(x_2)$ consists of two plane sections separated by a discontinuity along the $x_1$-axis). In this case, the equivalent loss landscape that the RGS algorithm is optimizing is strongly dependent on the optimization path and the starting point and therefore cannot be defined.

## C Ablation Studies

### C.1 Population Size

While PGPE recovers Gaussian Loss Smoothing in expectation, the quality of the gradient approximation depends strongly on the population size $n_{ps}$. In particular, a small population size $n_{ps}$ induces a high-variance estimate of the true smoothed loss, leading to noisy gradient estimates and thus slow learning or even stability issues. We illustrate this in Figure 8 where we show the loss surface along the gradient direction for different population sizes. We observe that for small population sizes the loss surface is indeed very noisy, only becoming visually smooth at $n_{ps} = 512$. Additionally, PGPE computes a gradient approximation in an $\frac{n_{ps}}{2}$-dimensional subspace, thus further increasing gradient variance if $n_{ps}$ is (too) small compared to the number of network parameters.

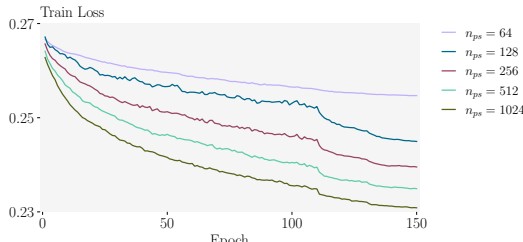

Table 5: Effect of the population size $n_{ps}$ on accuracy and training time with PGPE + DEEPPOLY training on `CNN3`.

| Popsize | Nat. [%] | Cert. [%] | GPU h |
|---|---|---|---|
| Init | 97.14 | 94.02 | - |
| 64 | 97.22 | 94.07 | 88 |
| 128 | 97.22 | 94.13 | 160 |
| 256 | 97.30 | 94.19 | 304 |
| 512 | 97.27 | 94.22 | 596 |
| 1024 | 97.43 | 94.50 | 1192 |

Figure 9: Evolution of Train Loss during training with different values for popsize $n_{ps}$. Note that for $n_{ps} = 64$ we trained with a lower learning rate because the value used in the other settings would make training unstable.

To assess the effect this has on the performance of PGPE training, we train the same `CNN3` on MNIST using population sizes between 64 and 1024, presenting results in Table 5. We observe that performance does indeed improve significantly with increasing population sizes (note the relative performance compared to initialization). This becomes even more pronounced when considering the training dynamics (see Figure 9). Unfortunately, the computational cost of PGPE is significant and scales linearly in the population size. We thus choose $n_{ps} = 256$ for all of our main experiments, as this already leads to training times of more than 1 week on 8 L4 GPUs for some experiments.

**Train Dynamics when varying population size** In Figure 9 we present the evolution of the Training Loss during training with different values for popsize $n_{ps}$. We observe significantly slower training as we decrease $n_{ps}$, confirming the theoretical prediction that using lower popsize decreases the quality of gradient estimations due to increased variance in the loss-sampling process.

### C.2 Standard Deviation

The standard deviation $\sigma$ used for Gaussian Loss Smoothing has a significant impact on the resulting loss surface as we illustrated in Figure 4 and discussed in §3. If $\sigma$ is chosen too small, the loss surface will still exhibit high sensitivity and gradients will only be meaningful very locally as discontinuities are barely smoothed. On the other hand, if $\sigma$ is chosen too large, the loss surface will become very flat and uninformative, preventing us from finding good solutions.

When estimating the smoothed loss in PGPE via sampling at moderate population sizes $n_{ps}$, the standard deviation $\sigma_{\mathrm{PGPE}}$ additionally affects the variance of the loss and thus gradient estimate. We illustrate this in Figure 10, where we not only see the increasing large-scale smoothing effect discussed above but also an increasing level of small-scale noise induced by a large $\sigma_{\mathrm{PGPE}}$ relative to the chosen population sizes $n_{ps}$.

To assess the effect this practically has on PGPE training, we train for 50 epochs with different standard deviations $\sigma_{\mathrm{PGPE}}$ and present the results in Figure 11. As expected, we clearly observe that both too small and too large standard deviations lead to poor performance. However, and perhaps surprisingly, we find that

training performance is relatively insensitive to the exact standard deviation as long as we are in the right order of magnitude between $10^{-3}$ and $10^{-2}$.

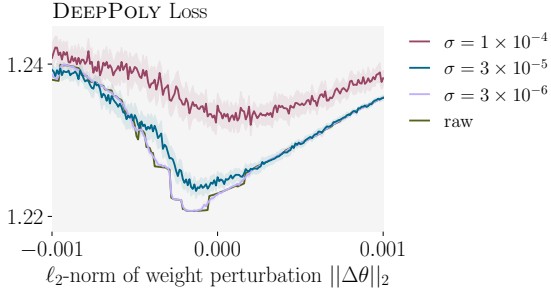

Figure 10: Effect of the standard deviation $\sigma_{\mathrm{PGPE}}$ on the induced loss surface in PGPE at a small population sizes of $n_{ps} = 32$.

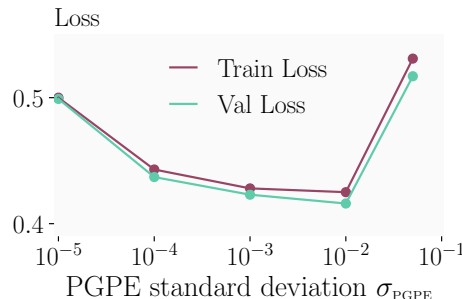

Figure 11: Train and Validation Losses after 50 epochs of training for different values of $\sigma_{\mathrm{PGPE}}$.

### C.3 Train Dynamics Comparison

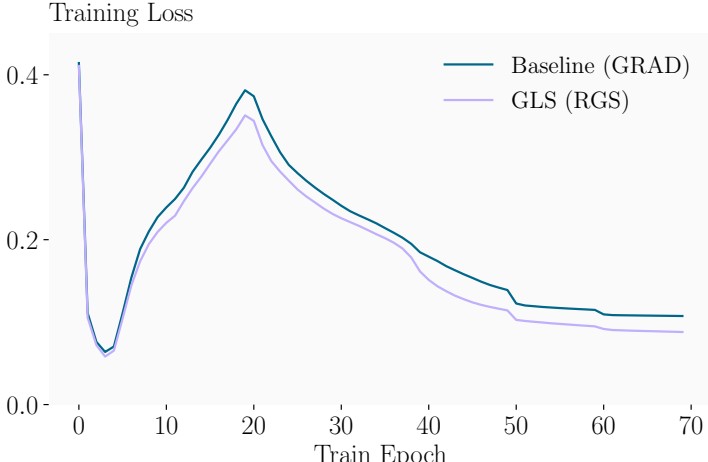

Figure 12: Evolution of training loss when training a `CNN5` on MNIST $\epsilon = 0.1$ using the DeepPoly convex relaxation with standard backpropagation (GRAD) and with GLS-based RGS.

In Figure 12 we show the differences between the training dynamics of GRAD-DeepPoly (Baseline) and RGS-DeepPoly for our CNN5 models trained on MNIST $\epsilon = 0.1$. We can observe that using RGS we obtain lower losses even from the $\epsilon$-annealing phase (first 20 epochs), resulting in a better training quality and better model after training is done.

## D   Additional Experimental Data

### D.1   Training CROWN-IBP-RGS on `CNN7`

While DeepPoly-RGS is too computationally extensive for scaling to `CNN7`, we can use RGS in combination with CROWN-IBP to prove that the advantages of GLS scale even to SOTA architectures. We present the results of training `CNN7` with CROWN-IBP-RGS in Table 6. We observe that RGS significantly increases the performance of CROWN-IBP when applied on `CNN7` without BatchNorm layers, but the improvement is less pronounced when using BatchNorm layers.

In order to accomodate the use of BatchNorm layers with RGS, we compute the estimated gradients by using the BatchNorm statistics independently for each sample of perturbed weights. To obtain the test statistics, we

reset the running stats to the population statistics for the mean trained network after each epoch, following the advice of Mao et al. (2024). While this approach is the most straightforward, it might not be the most effective way to use BatchNorm layers with RGS, and we leave the exploration of more sophisticated methods for future work.

Table 6: Comparison between `CNN7` networks trained with CROWN-IBP-RGS and SOTA GRAD methods on small perturbation settings

| Dataset | Network | Method | Nat. Acc. [%] | Cert. Acc. [%] | Adv. Acc. [%] |
|---|---|---|---|---|---|
| MNIST $\epsilon_\infty = 0.1$ | CNN7 with BN | IBP | 98.87 | 98.26 | 98.27 |
| | | CROWN-IBP | 99.10 | 98.13 | 98.22 |
| | | CROWN-IBP-RGS | 99.11 | 98.05 | 98.09 |
| | | TAPS | 99.16 | 98.52 | 98.58 |
| | CNN7 no BN | IBP | 98.50 | 97.40 | 97.42 |
| | | CROWN-IBP | 98.83 | 97.94 | 97.94 |
| | | CROWN-IBP-RGS | 99.19 | 98.09 | 98.18 |
| CIFAR-10 $\epsilon_\infty = 2/255$ | CNN7 with BN | IBP | 67.49 | 55.99 | 56.10 |
| | | CROWN-IBP | 70.90 | 58.80 | 59.93 |
| | | CROWN-IBP-RGS | 70.82 | 59.04 | 60.19 |
| | | MTL-IBP | 78.82 | 64.41 | 67.69 |
| | CNN7 no BN | IBP | 63.13 | 52.09 | 52.27 |
| | | CROWN-IBP | 67.82 | 55.36 | 56.62 |
| | | CROWN-IBP-RGS | 68.46 | 56.30 | 57.37 |

Finally, the results showcase that the promise of using GLS for certified training with tighter relaxations also scales to SOTA architectures.

### D.2 Comparison between RGS and PGPE on `CNN3` and `CNN3-tiny`

In Table 7 we provide additional experimental data comparing the performance of PGPE and RGS on `CNN3` and `CNN3-tiny`. In the case of `CNN3`, we observe that RGS and PGPE obtain similar performance on MNIST 0.1, but RGS is actually significantly better on CIFAR 2/255 (note that both are better than IBP and DeepPoly trained with Adam). We hypothesize that this might be due to: (1) on CIFAR-10 the `CNN3` network has more parameters than on MNIST ( 5k vs 7k due to different input sizes), thus the parameter space is larger; (2) we train for the same number of epochs on both datasets with PGPE, but standard certified training has shown networks on CIFAR-10 to converge slower than on MNIST. As a result, since PGPE trains slower due to the low-rank gradient problem, CIFAR-10 makes this worse, and insufficient training outweighs the theoretical benefit. These claims are further supported by our results for `CNN3-tiny` trained on MNIST where we observe that PGPE is significantly better than RGS, likely due to the smaller parameter space and faster convergence of the training.

### D.3 Comparison of GLS and Sharpness-Aware Minimization (SAM)

The gradient computation by perturbing the network weights used in PGPE and RGS has some similarities with the Sharpness-Aware Minimization (SAM, Foret et al. (2020)) algorithm. However, the SAM algotithm is fundamentally different to GLS. This is because GLS takes the expectation of neighborhood loss rather than the worst case loss; in fact, SAM is closer to adversarial training with FGSM (Goodfellow et al., 2015) rather than GLS. In particular, SAM does not resolve the discontinuity problem, while GLS provably solves it (Theorem 3.1). To see this, consider the threshold function $I(x > 0)$ and an initial $x_0 = 0.1$. Any single-point gradient based methods (including SAM) will only get zero gradient, and thus cannot optimize it. Therefore, while it is likely that GLS has the benefit of reduced sharpness as well, GLS enjoys fundamentally different benefits to SAM.

Table 7: Comparison of PGPE and RGS on `CNN3`

| Dataset | Network (params.) | Method | Nat. [%] | Cert. [%] | Adv. [%] |
|---|---|---|---|---|---|
| MNIST $\epsilon_\infty = 0.1$ | `CNN3-tiny` (1.1k) | IBP-GRAD | 89.76 | 82.46 | 82.48 |
| | | DEEPPOLY-GRAD | 89.24 | 68.47 | 68.57 |
| | | DEEPPOLY-PGPE | **91.94** | **85.00** | **85.04** |
| | | DEEPPOLY-RGS | 91.33 | 82.66 | 82.68 |
| MNIST $\epsilon_\infty = 0.1$ | `CNN3` (5.2k) | IBP-GRAD | 96.02 | 91.23 | 91.23 |
| | | DEEPPOLY-GRAD | 95.95 | 90.04 | 90.08 |
| | | DEEPPOLY-PGPE | **97.44** | 91.53 | 91.79 |
| | | DEEPPOLY-RGS | 97.37 | **91.88** | **92.03** |
| CIFAR-10 $\epsilon_\infty = 2/255$ | `CNN3` (6.8k) | IBP-GRAD | 48.05 | 37.69 | 37.70 |
| | | DEEPPOLY-GRAD | 47.70 | 36.72 | 36.72 |
| | | DEEPPOLY-PGPE | 54.17 | 38.95 | 40.20 |
| | | DEEPPOLY-RGS | **54.93** | **41.14** | **42.03** |

To confirm this empirically, we apply SAM to IBP and DEEPPOLY training. Specifically, we update the parameters with gradients computed based on the adversarially perturbed network $w' = w + \rho \times \nabla_w L / \|\nabla_w L\|_2$. We train with IBP and DEEPPOLY on MNIST $\epsilon = 0.1$ with the same CNN3 architecture used in the paper. The results are shown in Table 8, all networks certified with MN-BaB.

Table 8: Comparison of GLS methods with SAM. `CNN3` networks trained on MNIST $\epsilon = 0.1$.

| Method | Nat. [%] | Cert. [%] |
|---|---|---|
| IBP-GRAD | 96.02 | 91.23 |
| IBP-SAM $\rho = 0.1$ | 96.08 | 90.20 |
| IBP-SAM $\rho = 0.01$ | 96.32 | 93.32 |
| IBP-SAM $\rho = 0.001$ | 95.80 | 91.73 |
| DEEPPOLY-GRAD | 95.95 | 90.04 |
| DEEPPOLY-SAM $\rho = 0.1$ | 94.22 | 88.39 |
| DEEPPOLY-SAM $\rho = 0.01$ | 96.93 | 92.34 |
| DEEPPOLY-SAM $\rho = 0.001$ | 96.95 | 90.91 |
| DEEPPOLY-PGPE | 97.44 | 91.53 |
| DEEPPOLY-RGS | 97.37 | 91.88 |

We observe that for a correctly chosen hyperparameter ($\rho = 0.01$), SAM does indeed improve performance for IBP and DP. While SAM performs better than PGPE for this very shallow network, as expected from our previous theoretical analysis, it does not address the paradox. In particular, IBP-SAM still performs better than DP-SAM uniformly for every choice of $\rho$. While combining SAM with PGPE or other certified training methods might thus constitute an interesting future direction, it does not explain the reranking of approximation methods (DEEPPOLY-PGPE > IBP-PGPE vs IBP-SAM > DEEPPOLY-SAM) we observe for PGPE. We therefore conclude that the sharpness aware aspect of PGPE is not (solely) responsible for its effectiveness in resolving the paradox of certified training.

# E  Additional Discussion

## E.1  Discussion on the regularization induced by IBP for large perturbations

In §4.4 we show that RGS improves the performance of tight relaxations surpassing the SOTA methods on the same architecture for the setting of small perturbations. However, this improvement is not as substantial in the case of large perturbations. Based on our experimental intuitions and the findings of prior works (Mao et al., 2024; 2023b), we hypothesize that the regularization induced by training with IBP bounds boosts the network's certifiability in the case of large perturbations. We note that the $L_1$ regularization has been

sufficiently tuned by Mao et al. (2024) for MTL-IBP, thus only increasing $L_1$ regularization strength cannot achieve the kind of regularization needed for certifiability. Therefore, we speculate that large $\epsilon$ leads to much more unstable neurons, leading to exponential growth of certification difficulty. Thus, for large $\epsilon$, strong (and maybe unnecessary to robustness) regularization is required to further reduce certification difficulty. Note that the effects of this IBP regularization are more complex than just limiting the magnitude of the weights, as described by Mao et al. (2023b).

### E.2 Discussion on the relation between GLS and Randomized Smoothing

While Gaussian Loss Smoothing (GLS) and Randomized Smoothing (RS) (Cohen et al., 2019) both involve Gaussian noise, they differ fundamentally in purpose and mechanism. RS applies smoothing in the input space, typically training on noisy inputs to enable probabilistic certification at inference time. This can be seen as a form of data augmentation and requires specialized training tailored to the target noise level.

In contrast, GLS applies smoothing in the parameter (weight) space, modifying the loss landscape to improve optimization—especially in the presence of discontinuities introduced by tight convex relaxations. As such, GLS is more closely related to classical smoothing methods used for gradient estimation and optimization in non-smooth settings.

Importantly, the certifications provided by RS are probabilistic and require substantial runtime sampling (e.g., 100+ forward passes per input), whereas our method yields deterministic certificates based on convex bounds. Although GLS-based training can be more computationally intensive during optimization, inference is straightforward and efficient, requiring no changes to the trained model.

## F Additional Training Details

### F.1 Standard Certified Training

We train with the Adam optimizer (Kingma & Ba, 2015) with a starting learning rate of $5 \times 10^{-5}$ for 70 epochs on MNIST and 160 epochs on CIFAR-10 and TINYIMAGENET. We use the first 20 epochs on MNIST and 80 epochs on CIFAR-10 and TINYIMAGENET for $\epsilon$-annealing, with the first epoch having $\epsilon = 0$ for CIFAR-10 and TINYIMAGENET. We decay the learning rate by a factor of 0.2 after epochs 50 and 60 for MNIST and respectively 120 and 140 for CIFAR-10 and TINYIMAGENET. For certified training on MNIST and CIFAR-10, we use the IBP initialization proposed by Shi et al. (2021). For PGD training and for certified training on TINYIMAGENET we use the Kaiming uniform initialization (He et al., 2015).

### F.2 PGPE Training

We use a training schedule of 150 epochs, with a batch size of 512 for MNIST and 128 for CIFAR-10. We train with a starting learning rate of 0.0003 and we decay it twice by a factor of 0.4 after the $110^{\text{th}}$ and $130^{\text{th}}$ epoch. We use the first 50 epochs for $\epsilon$-annealing only when training with the large value of $\epsilon$ for each dataset (MNIST $\epsilon = 0.3$ and CIFAR-10 $\epsilon = 8/255$). Due to time constraints, we start all training rounds from models trained with the PGD loss in a standard gradient-based setting.

**Training with non-differentiable bounding methods** In addition, for training with $\alpha$-CROWN + PGPE, we use the same training schedule and hyperparameters as for standard PGPE training. For the slope optimization procedure of $\alpha$-CROWN, we initialize all slopes with the value of 0.5 and we conduct only one optimization step with step size 0.5 for each batch, resulting in all slopes having a value of either 0.0 or 1.0. In this way, we obtain a boost in tightness when compared to standard DeepPoly, while increasing the computational cost only by a factor of 2. Slope optimization with multiple steps and smaller step sizes can further increase the tightness of the relaxation, but at the cost of increased computational complexity.

### F.3 RGS Training

We use the same training schedules and hyperparameters as Standard Certified Training. In addition, we use a population size of $n_{ps} = 2$ for all experiments, and an initial standard deviation of $\sigma_{\text{RGS}} = 10^{-3}$ for all experiments. We decay the standard deviation used for sampling gradients by a factor of 0.4 at the same training steps as the learning rate. We use the same initialization schemes as for standard certified training, unless specified otherwise.

### F.4 Architectures

In Table 9 we present the network architectures used for all our experiments.

Table 9: Network architectures of the convolutional networks for CIFAR-10 and MNIST. All layers listed below are followed by a ReLU activation layer. The output layer is omitted. 'Conv c h×w/s/p' corresponds to a 2D convolution with c output channels, an h×w kernel size, a stride of s in both dimensions and an all-around zero padding of p.

| CNN3-tiny | CNN3 | CNN5 | CNN5-L |
|---|---|---|---|
| Conv 2 5×5/2/2 | Conv 8 5×5/2/2 | Conv 16 5×5/2/2 | Conv 64 5×5/2/2 |
| Conv 2 4×4/2/1 | Conv 8 4×4/2/1 | Conv 16 4×4/2/1 | Conv 64 4×4/2/1 |
| | | Conv 32 4×4/2/1 | Conv 128 4×4/2/1 |
| | | FC 512 | FC 512 |

### F.5 Dataset and Augmentation

We use the MNIST (LeCun et al., 2010), CIFAR-10 (Krizhevsky et al., 2009) and TinyImageNet (Le & Yang, 2015) datasets for our experiments. All are open-source and freely available with unspecified license. The data preprocessing mostly follows De Palma et al. (2024). For MNIST, we do not apply any preprocessing. For CIFAR-10 and TinyImageNet, we normalize with the dataset mean and standard deviation and augment with random horizontal flips. We apply random cropping to $32 \times 32$ after applying a 2-pixel zero padding at every margin for CIFAR-10, and random cropping to $64 \times 64$ after applying a 4-pixel zero padding at every margin for TinyImageNet. We train on the corresponding train set and certify on the validation set, as adopted in the literature (Shi et al., 2021; Müller et al., 2023; Mao et al., 2023a; De Palma et al., 2024).

### F.6 Training costs (Time and Resources)

**Theoretical Costs of GLS** The theoretical costs of any GLS-based method scales linearly with the population size $n_{ps}$ used to obtain gradient estimates. In the case of PGPE, we need to use a large population size (e.g. $n_{ps} = 256$), which makes the algorithm very costly: we need to compute the loss via forward pass $n_{ps}$ times. Estimating the gradient after computing the losses is negligible, so the total time complexity of one epoch is $O(n_{ps} * F)$, where $F$ would be the time taken by the standard certified training algorithm to only compute the forward pass in one epoch. For RGS, there's no need to use large population sizes ($n_{ps} = 2$ works well enough), so the total training time in our experiments is just $n_{ps} = 2$ times larger than normal certified training (includes both forward and backward passes because we use backpropagation to obtain gradients).

**Experimental Costs** For PGPE and RGS training, we used between 2 and 8 NVIDIA L4-24GB or NVIDIA A100-40GB GPUs. For standard certified training and for certification of all models we used single L4 GPUs.

In Table 10 we present a detailed analysis of the training costs of the PGPE and RGS methods for all of our experimental settings (Note that the cost of DeepPoly-PGPE for CNN5 was estimated based on training for only 1 epoch). In Table 11 we present the training costs for the baseline standard certified training methods for comparison.

Table 10: Training costs and workload distribution across GPUs / actors for each train setting.

| Datset | Network | Method | GPUs | Num. Actors | Time/epoch (min) | GPU-h/ epoch |
|---|---|---|---|---|---|---|
| MNIST | CNN3-tiny | DEEPPOLY-PGPE | 4 x L4 | 4 | 25 | 1.73 |
| | | $\alpha$CROWN-PGPE | 8 x L4 | 8 | 44 | 5.86 |
| | CNN3 | IBP-PGPE | 2 x L4 | 4 | 2.8 | 0.09 |
| | | CROWN-IBP-PGPE | 2 x L4 | 4 | 8.5 | 0.28 |
| | | HBOX-PGPE | 8 x L4 | 8 | 31 | 4.13 |
| | | DEEPPOLY-PGPE | 8 x L4 | 8 | 27 | 3.60 |
| | CNN5 | DEEPPOLY-PGPE (est.) | 8 x L4 | 8 | $\approx 300$ | $\approx 40$ |
| | | DEEPPOLY-RGS | 8 x L4 | 8 | 7.5 | 1 |
| | CNN5-L | DEEPPOLY-RGS | 8 x A100 | 8 | 35 | 4.68 |
| CIFAR-10 | CNN3 | IBP-PGPE | 2 x L4 | 4 | 6.9 | 0.23 |
| | | CROWN-IBP-PGPE | 4 x L4 | 8 | 8.5 | 0.57 |
| | | DEEPPOLY-PGPE | 8 x L4 | 8 | 42 | 5.6 |
| | CNN5 | DEEPPOLY-PGPE (est.) | 8 x L4 | 8 | $\approx 360$ | $\approx 48$ |
| | | DEEPPOLY-RGS | 8 x L4 | 8 | 16 | 2.2 |
| | CNN5-L | DEEPPOLY-RGS | 8 x A100 | 8 | 33 | 4.4 |
| TINYIMAGENET | CNN5 | DEEPPOLY-RGS | 8 x A100 | 8 | 41 | 5.5 |

Table 11: Training times of CNN5 on 1xL4 GPU with standard autograd training depending on training method.

| Dataset | Method | Train Time (1xL4 gpu) |
|---|---|---|
| MNIST | PGD | 15m |
| | IBP | 10m |
| | SABR | 20m |
| | STAPS | 25m |
| | MTL-IBP | 40m |
| CIFAR-10 | PGD | 1h30m |
| | IBP | 1h00m |
| | SABR | 2h00m |
| | STAPS | 2h30m |
| | MTL-IBP | 3h10m |
| TINYIMAGENET | PGD | 3h15m |
| | IBP | 2h20m |
| | MTL-IBP | 4h20m |

