# OpenReview forum: "Gaussian Loss Smoothing Enables Certified Training with Tight Convex Relaxations"
_TMLR — Accepted by TMLR_

### Review · Reviewer_TWPN · 2025-04-09

**Summary Of Contributions:**

This paper addresses the challenge of training neural networks with high certified accuracy against adversarial examples. It notes that while tight convex relaxations are beneficial for certification, they can surprisingly underperform looser relaxations during training. Previously researchers hypothesized this is due to the loss surface's discontinuity, non-smoothness, and perturbation sensitivity caused by tighter relaxations.

To mitigate these issues, the paper theoretically proposes and empirically validates Gaussian Loss Smoothing (GLS). They introduce two GLS-based training methods: PGPE (zeroth-order, for non-differentiable relaxations) and RGS (first-order, more efficient). Extensive experiments demonstrate that these methods, when combined with tight relaxations, outperform state-of-the-art techniques on the same network architecture across various settings. The findings highlight the potential of GLS for training certifiably robust neural networks and suggest a direction for effectively utilizing tighter relaxations in certified training.

**Audience:**

Yes

**Claims And Evidence:**

Yes

**Requested Changes:**

1) The table 1 seems misleading. It marked the original gradient based methods as red, and marked theirs as blue. It makes the reader feel it has been always improved, but it seems like the certified accuracy gets lower for IBP, and they still marked it as blue to make it feel improved. I hope the authors change the design of the table a bit.

2) Has there really been no prior approach that utilized smoothing by adding noise? Looking at the authors' paper, it seems like a relatively old field, dating back to around 2018, and smoothing by adding noise to polish the landscape is a fairly common technique in the machine learning community. It's quite interesting that there haven't been attempts to combine these. Could you please conduct a more thorough investigation and provide a confirmation? That would greatly help with the decision. Also, could you include related works on smoothing?

For example, compared to 'Certified Adversarial Robustness via Randomized Smoothing', what is the main difference?

**Strengths And Weaknesses:**

1) Strong theoretical findings (Theorem 3.1), solved the issues found by the works of the previous researchers.
2) Empirical Confirmation through various settings.

Even though I am not an expert in this field, according to their statements, this paper seems to have great potential in machine learning society. Even though I don't have strong confidence in my decision, I'd like to lean toward acceptance.

---

> ### Author Response · Authors · 2025-05-07
> **Response to Reviewer $\RT$**
>
> We are happy to hear that Reviewer $\RT$ considers our studied problem important, the proposed method well-motivated by theoretical findings, and the empirical results varied and convincing. In the following, we address all concrete questions raised by Reviewer $\RT$. We will incorporate reviewer’s writing suggestions in our revised manuscript.
>
> **Q1: Colors used in Figure 1 seem misleading (red, blue). Could the authors clarify the meaning of this figure?**
>
> We thank the reviewer for pointing out the potential confusion caused by the use of color in Figure 1. The red and blue color shading in the figure is not meant to imply a universal notion of “good” or “bad” performance. Instead, the colors are used independently within each column to indicate relative values—lighter shades for lower values and darker shades for higher values.
>
> Our intention was to emphasize that under GRAD training, the highest certified accuracy is achieved using the loosest relaxation (IBP), which contradicts theoretical expectations—this is what we refer to as the “paradox” of certified training. In contrast, GLS-based methods (shown in blue) produce certified accuracies that correlate positively with relaxation tightness, aligning better with theoretical intuition.
>
> To avoid this misunderstanding, we have revised the caption in the updated manuscript to clarify the role of the colors and the purpose of the figure. We hope this addresses the reviewer’s concern.
>
> **Q2: Are there any related works that study noise-based smoothing in the field of deterministic certified training?**
>
> To the best of our knowledge, this is the first work to introduce noise-based smoothing in the context of deterministic certified training. While the field of certified training has seen extensive development—especially around Interval Bound Propagation (IBP)—prior work has not explored smoothing the loss surface as a solution. For example, Shi et al. (2021) [1] proposed a new initialization technique, architectural changes and a regularization strategy to improve IBP-based training, but did not consider loss smoothing. This omission is understandable, as IBP’s loose relaxation tends to yield smooth optimization landscapes and does not suffer from the discontinuity or sensitivity issues observed with tighter relaxations.
>
> Our motivation stems from the findings of Jovanovic et al. (2022) [2], who demonstrated that tighter convex relaxations lead to highly sensitive and discontinuous loss surfaces, making deterministic training unstable and less effective. In this work, we show both theoretically and empirically that Gaussian Loss Smoothing significantly mitigates these issues—thereby enabling stable and effective certified training with tight relaxations like DeepPoly. Additionally, our experiments confirm that smoothing provides little to no benefit when applied to IBP, further supporting this interpretation.
>
> **Q3: How does your method relate to Randomized Smoothing [3]?**
>
> While Gaussian Loss Smoothing (GLS) and Randomized Smoothing (RS) both involve Gaussian noise, they differ fundamentally in purpose and mechanism. RS applies smoothing in the input space, typically training on noisy inputs to enable probabilistic certification at inference time. This can be seen as a form of data augmentation and requires specialized training tailored to the target noise level.
>
> In contrast, GLS applies smoothing in the parameter (weight) space, modifying the loss landscape to improve optimization—especially in the presence of discontinuities introduced by tight convex relaxations. As such, GLS is more closely related to classical smoothing methods used for gradient estimation and optimization in non-smooth settings.
>
> Importantly, the certifications provided by RS are probabilistic and require substantial runtime sampling (e.g., 100+ forward passes per input), whereas our method yields deterministic certificates based on convex bounds. Although GLS-based training can be more computationally intensive during optimization, inference is straightforward and efficient, requiring no changes to the trained model.
>
> We have added this extended discussion in the revised manuscript as App. E.2.
>
> **References**
>
> [1] Shi et al., Fast Certified Robust Training with Short Warmup, NeurIPS 2021
>
> [2] Jovanovic et al., On the paradox of certified training, TMLR 2022
>
> [3] Cohen et al., Certified Adversarial Robustness via Randomized Smoothing, ICML 2019

---

### Review · Reviewer_h2Ty · 2025-04-22

**Summary Of Contributions:**

The paper addresses what it calls the Paradox of Certified Training, where certified robust training of neural networks on tighter (i.e. theoretically better) convex relaxations often leads to empirically worse performance. It identifies three key properties that cause this poor performance: discontinuities, non-smoothness, and sensitivity in the tighter relaxations. It aims to solve these by using Gaussian Loss Smoothing (GLS), i.e. applying a Gaussian kernel to the loss.

The overall contributions are four-fold: theoretical analysis of GLS, a zeroth-order training method based on PGPE with GLS, a first-order training method based on RGS with GLS, and evaluation results comparing to baselines.

The paper goes in detail through preliminaries about training for certified robustness, then explanation and investigation of GLS itself and how it handles discontinuities, non-smoothness (Lipschitz continuity), and sensitivity. It then lays out results comparing convex relaxations of varying tightness with and without GLS on multiple datasets and with both optimization methods. It primarily shows performance that outdoes loose relaxations, with less empirical investigation into the three properties.

**Audience:**

Yes

**Claims And Evidence:**

Yes

**Requested Changes:**

See weaknesses

**Strengths And Weaknesses:**

## Quality
### Strengths
- Well-designed project. Begins with a clear idea, provides framework, does theoretical analysis, and then does experimentation. The theoretical analysis maps very nicely onto discontinuity, non-smoothness, and sensitivity as defined.
- Experimental design is very thorough. I particularly appreciate the inclusion of adversarial accuracy, which feels honest and thorough
- Results show improvement over baselines
- Good discussion comparing the results, though it could be more structured
### Weaknesses
- Would like to see more visualization/intuition built about what the empirical results tell us about the three properties
- Could be more upfront about the novelty of the paper in particular. From what I understand it is novel, but at times it's difficult to distinguish the paper's claims and the prior work, particularly wrt PGPE and RGS. Once we go through the longer subsections in Sec 3 it's clearer, but earlier language could be toned down.

## Clarity
### Strengths
- Overall very well-written. I do not have a deep theoretical ML background, so while I limited in my ability to critically evaluate this paper's situation in existing literature or the soundness of its framework, it does make its own framework intuitive and easy to understand
- Results section has useful structure, being claims-driven and signposted
- There's a lot of theory, but I found it possible to follow. The figures and explanations really help build intuition and visualize, which is crucial
### Weaknesses
- Fig. 5 should be visualized differently, as it feels crucial for building intuition but is hard to understand
- Results section, though well-signposted, is somewhat verbose. It becomes hard to keep everything straight. With a few reads it's not so bad, but it may help to ease the burden on less-informed readers by calling out key points more.

---

> ### Author Response · Authors · 2025-05-07
> **Response to Reviewer $\Rh$**
>
> We are happy to hear that Reviewer $\Rh$ considers our work well structured, our theoretical analysis clear, and our experimental results thorough and convincing. We are particularly encouraged by the positive opinions from Reviewer $\Rh$. In the following, we address all concrete questions raised by Reviewer $\Rh$.
>
> **Q1: Could the authors provide more visualization/intuition built about what the empirical results tell us about the three properties?**
>
> Sure. To provide further insights, we have added a new figure (now numbered as Fig 5) in the revised manuscript. This figure presents the evolution of some scores we used as proxies for measuring the three undesirable properties of the loss landscape as we increase the smoothing strength for the same CNN3 network and settings used in Figure 4. Namely, we compute the maximum magnitude of finite differences (defined as $\Delta f(x)/\Delta x$) as an approximation for the magnitude of the *Discontinuities* in the loss function, the average curvature of the loss function (defined as the average magnitude of the 2nd order derivative) as a measure of *Non-Smoothness*, and the deviation from convexity (as defined in Section 3.1) as a proxy for *Sensitivity*. We observe that smoothing significantly reduces all of these scores for DeepPoly and CROWN-IBP, therefore mitigating the optimization issues imposed by the three undesirable properties.
>
> **Q2: At times it is difficult to distinguish the paper's claims and the prior work, particularly wrt PGPE and RGS. Once we go through the longer subsections in Sec 3 it's clearer, but earlier language could be toned down.**
>
> We have slightly rephrased our Introduction in the revised manuscript (paragraphs **This Work** and **Main Contributions**) to make it clearer that PGPE and RGS are existing algorithms that we use to augment certified training resulting in better performance when training with Tight Convex Relaxations. If the reviewer has further suggestions, we are happy to revise our writing.
>
> **Q3: Could the authors redesign Fig. 5?**
>
> Sure. We have reworked the figure (now re-numbered as Figure 6 in the revised manuscript) and its caption to better highlight the steps of the PGPE algorithm.
>
> **Q4: Results section, though well-signposted, is somewhat verbose. It becomes hard to keep everything straight. With a few reads it's not so bad, but it may help to ease the burden on less-informed readers by calling out key points more.**
>
> We have restructured our Experimental Results section in the revised manuscript to make the presented experiments and conclusions easier to follow. If the reviewer has further suggestions, we are happy to revise our writing.

---

### Review · Reviewer_GxFb · 2025-04-29

**Summary Of Contributions:**

This paper considers addresses the challenges associated with training neural networks using tight convex relaxations for certified robustness. It is known that the discontinuity, non-smoothness, and sensitivity of the loss surface are key factors in the performance of the certified training. The authors propose Gaussian Loss Smoothing (GLS) as a general framework to mitigate these issues. The contributions include
1. they provide theoretical justification for how GLS can smooth the loss function and reduces the deviation from convexity.
2. Algorithms: proposed PGPE and RGS implmenetations of GLS.
3. Empirical evaluations: They compared the GLS with other baseline methods on a variety of datasets and models (although the general architecture is limited to CNN)

Overall, the paper demonstrates the potential of GLS as an effective enhancement to existing convex relaxation-based training methods.

**Audience:**

Yes

**Broader Impact Concerns:**

No concern

**Claims And Evidence:**

Yes

**Requested Changes:**

There are few suggestions and questions that I appreciate it if the author could address them:

**Questions**:
1. When defining the $\epsilon$-neighborhood, what norm (i.e., what value of $p$) is used in the paper?
2. In the equation immediately following (1), define the index $i$.
3. Table 1 Terminology: Please clarify the $\epsilon_{\infty}$. Also, instead of abbreviations "Nat.", "Cert.", and "Adv." I recommend the authors to use the full words.
4. Is GRAD essentially the baseline method, i.e., it applies the relaxation without GLS as in their original papers?
5. What is the added computational complexity and training overhead of incorporating GLS?
6. How does GLS influence the convergence rate of the training process? It would be helpful if the authors could include training loss curves with and without GLS for one model, to compare convergence dynamics.

**Minor Suggestions**:
1. The current title suggests that certified training is enabled by the proposed method. However, since convex relaxations already allow for certified training, and GLS improves their effectiveness, a term like *Improves* or *Enhances* may better reflect the contribution than *Enables*.
2. It would help readers if the authors explicitly state that GLS is not an independent relaxation technique, but rather complements existing convex relaxations to enhance their performance.
3. Introducing a notation table or a brief subsection summarizing the key mathematical symbols.
4. In the second paragraph of Section 3.3, the first sentence could be rephrased to clarify that PGPE samples *n* weight perturbations.

**Strengths And Weaknesses:**

**Strengths:**

GLS is well-motivated and the *general* ideas are presented clearly. The authors have provided a theoretical foundation for GLS and evaluated the performance empirically across multiple benchmarks.

**Weaknesses:**

- Certain parts of the paper (especially technical explanations and notations) could be written more clearly. (please see suggestions)
- Outdated Literature:The paper does not reference publications from 2024 or 2025, making the paper appear somewhat dated. Mentioning  recent developments would better situate the contribution within the current state of the field.

---

> ### Author Response · Authors · 2025-05-07
> **Response to Reviewer $\RG$ (1/2)**
>
> We are happy to hear that Reviewer $\RG$ considers our proposed method well-motivated, the empirical results strong and varied, and the paper well-written. In the following, we address all concrete questions raised by Reviewer $\RG$.
>
> **Q1:Could the authors provide a discussion about more recent related publications (2024-2025)?**
>
> To the best of our knowledge, there has been relatively little new work in *deterministic certified training* over the past year. The most recent advances include Expressive Losses [1], TAPS [2], SABR [3], and CTBench [4], which consolidates recent methods into a unified evaluation framework and benchmark. There have also been some works focused on theoretical aspects of certified training and network expressivity [5,6,7].  All of these have already been discussed in our manuscript. If the reviewer considers that we should include discussions about other recent related works we might have missed, we are happy to do so.
>
> In the adjacent field of neural network verification, research has converged toward efficient and scalable branch-and-bound frameworks, such as MN-BAB [8] and OVAL-BAB [9], which represent the current state of the art in exact verifiers.
>
> **Q2: When defining the $\epsilon$-neighborhood, what norm (i.e., what value of p) is used in the paper?**
>
> Even though our definitions in Section 2.1 are valid in the general case of using any $L_p$ norm, throughout our work we only use $L_\infty$-bounded perturbations. We have clarified this in the revised manuscript (end of Sec. 2.1).
>
> **Q3: What does the index i represent in the equation immediately following (1)?**
>
> The index $i$ was supposed to represent the layer-wise fashion in which these bounds are usually computed. To avoid confusion with the index $i$ used to denote different classes in previous equations, we have replaced the index with $j$ and we have clarified this in the revised manuscript.
>
> **Q4: Table 1 Terminology: Please clarify the $\epsilon_\infty$. Also, instead of abbreviations "Nat.", "Cert.", and "Adv." I recommend the authors to use the full words.**
>
> In Table 1, $\epsilon_\infty$ represents the radius of the $L_\infty$ ball used as the perturbation space around each input for certification. We have clarified this and included reviewer’s suggestions in the revised manuscript.
>
> **Q5: Is GRAD essentially the baseline method, i.e., it applies the relaxation without GLS as in their original papers?**
>
> Yes. We have clarified this in the revised manuscript (in the captions of Fig 1 and Table 1).
>
> **Q6: What is the added computational complexity and training overhead of incorporating GLS?**
>
> The computational complexity and training overhead obtained by incorporating GLS depends on the specific algorithm used to instantiate GLS. In Section 3 we present and discuss two algorithms that can approximate GLS: PGPE and RGS. The training overhead of these methods depends on the population size used to obtain gradient estimates.
>
> In the case of PGPE, we need to use a large population size (e.g. $2n=256$), which makes the algorithm very costly: we need to compute the loss via forward pass $2n$ times. Estimating the gradient after computing the losses is negligible, so the total time complexity of one epoch is $O(2n\cdot F)$, where F would be the time taken by the standard certified training algorithm to only compute the forward pass in one epoch.
>
> For RGS, there’s no need to use large population sizes ($n=2$ works well enough), so the total training time in our experiments is just $n=2$ times larger than normal certified training.
>
> We have included these details in App. F.6 in the revised manuscript.
>
> **Q7: How does GLS influence the convergence rate of the training process? Could the authors include training loss curves with and without GLS for one model, to compare convergence dynamics?**
>
> Sure. We have added Figure 10 in Appendix C.3 to show the differences between the training dynamics of GRAD-DeepPoly (Baseline) and RGS-DeepPoly for our CNN5 models trained on MNIST $\epsilon=0.1$. We can observe that using RGS we obtain lower losses even from the $\epsilon$-annealing phase (first 20 epochs), resulting in better training dynamics and a better model after training is done.

---

> ### Author Response · Authors · 2025-05-07
> **Response to Reviewer $\RG$ (2/2)**
>
> **Q8: Given that certified training using Convex Relaxations is already used, is the word “Enables” in the title an overstatement?**
>
> We appreciate the reviewer’s observation. While it is true that certified training with convex relaxations has a long history, the vast majority of successful methods rely on loose relaxations such as IBP. These approaches heavily over-approximate the worst-case loss, resulting in high regularization and suboptimal trade-offs between natural and certified accuracy.
>
> Previous work by Jovanovic et al. [10] highlighted that training with tight convex relaxations, while theoretically appealing, suffers from practical optimization issues—specifically, discontinuity and sensitivity of the loss surface—which make training unstable and ineffective in practice.
>
> In this work, we show that Gaussian Loss Smoothing (GLS) addresses these challenges directly. Our method is the first, to our knowledge, to demonstrate that tight relaxations such as DeepPoly can consistently outperform IBP in certified accuracy across multiple settings, closing the long-standing gap between theoretical potential and empirical performance.
>
> Therefore, we believe the term “Enables” is appropriate: while tight relaxations have been studied before, our work is the first to make their use viable and effective in practice for deterministic certified training.
>
> **Q9: It would help readers if the authors explicitly state that GLS is not an independent relaxation technique, but rather complements existing convex relaxations to enhance their performance.**
>
> Sure. We have clarified this in the revised manuscript (see Introduction, **This Work** paragraph).
>
> **Q10: Could the authors introduce a notation table or a brief subsection summarizing the key mathematical symbols?**
>
> Sure. We have added a new section explaining notation as App. A in the revised manuscript.
>
> **References**
>
> [1] De Palma et al., Expressive Losses for Verified Robustness via Convex Combinations, ICLR 2024
>
> [2] Mao et al., Connecting Certified and Adversarial Training, NeurIPS 2023
>
> [3] Mueller et al., Certified Training: Small Boxes are All You Need, ICLR 2023
>
> [4] Mao et al., CTBENCH: A Library and Benchmark for Certified Training, arXiv 2024
>
> [5] Mao et al., Understanding Certified Training with Interval Bound Propagation, ICLR 2024
>
> [6] Baader et al., Expressivity of ReLU-Networks under Convex Relaxations, ICLR 2024
>
> [7] Mao et al., On the Expressiveness of Multi-Neuron Convex Relaxations, arXiv 2025
>
> [8] Ferrari et al., Complete verification via multi-neuron relaxation guided branch-and-bound, ICLR 2022
>
> [9] De Palma et al., Improved Branch and Bound for Neural Network Verification via Lagrangian Decomposition, arXiv 2021
>
> [10] Jovanovic et al., On the paradox of certified training, TMLR 2022

---

> ### Comment · Reviewer_GxFb · 2025-05-07
> **Thanks for the response**
>
> I greatly appreciate the authors' detailed responses. My concerns and questions have been fully addressed.

---

### Author Response · Authors · 2025-05-07
**General Response to Reviews**

We thank all reviewers for their insightful reviews, interesting questions, and helpful feedback. $\newcommand{\RG}{\textcolor{green}{GxFb}} \newcommand{\Rh}{\textcolor{blue}{h2Ty}} \newcommand{\RT}{\textcolor{orange}{TWPN}}$We are particularly encouraged that reviewers consider our work novel and well-motivated ($\RG$, $\Rh$, $\RT$), the experimental results and conclusions insightful for the community ($\RG$, $\Rh$, $\RT$), and the paper well-written ($\RG$, $\Rh$). We have incorporated all changes requested by the reviewers in the revised manuscript. Because we have included new figures and reworked some sections, the main text is now slightly longer than the 12 page threshold, but we consider these additions to strengthen our paper.

As we did not identify any shared questions among reviewers, we will address each reviewer’s questions in individual responses. We look forward to the reviewers’ replies.

---

### Decision · Action_Editor_xojZ · 2025-06-16

**Recommendation:** Accept as is

**Audience:**

Yes

**Audience Explanation:**

Training adversarially-robust machine learning models is a long-standing problem in modern day ML. As such, this paper will be of interest to a significant portion of the TMLR community.

**Claims And Evidence:**

Yes

**Claims Explanation:**

This paper addresses the question of improving certified robust training of neural networks, and in particular the issue that employing tighter (theoretically superior) convex relaxations often results in worse empirical performance compared to looser relaxations. To resolve this, the paper introduces a Gaussian Loss Smoothing (GLS) technique. The work includes some theoretical results (showing that the resulting loss is smooth and bounded), and performs empirical evaluation of two different GLS instantiations (PGPE and RGS), including extensive experimental comparisons.

All reviewers found the paper to be clearly motivated and well structured, addressing an important gap in the literature. Some references and discussions were missing in the first version of this paper, but the rebuttal stage has been productive and comments from reviewers have been successfully addressed. All reviews are leaning or recommending acceptance, and I concur with this decision.